# Subtle influence of AMOC on seasonal SST hindcast skill in the North Atlantic

Julianna C. Oliveira[1], Leonard F. Borchert[2,3], Aurélie Duchez[4], Mikhail Dobrynin[3,5], and Johanna Baehr[3]

[1]University of Southampton, Southampton, United Kingdom. *Present address:* Helmholtz-Zentrum Hereon, Geesthacht, Germany and International Max Planck Research School on Earth System Modelling, Max Planck Institute for Meteorology, Hamburg, Germany

[2]Sorbonne Universités (SU/CNRS/IRD/MNHN), LOCEAN Laboratory, Institut Pierre Simon Laplace (IPSL), Paris, France. *Present address:* École Normale Supérieure, LMD, IPSL, Paris, France

[3]Institute of Oceanography, Center for Earth System Research and Sustainability (CEN), Universität Hamburg, Hamburg, Germany

[4]ESAIP La Salle, Aix en Provence, France and National Oceanography Centre, Southampton, United Kingdom

[5]Deutscher Wetterdienst (DWD), Hamburg, Germany

**Correspondence:** Julianna C. Oliveira (julianna.carvalho@hereon.de)

**Abstract.** We investigate the impact of the strength of the Atlantic Meridional Overturning Circulation (AMOC) at 26°N on the prediction of North Atlantic sea surface temperature anomalies (SSTA) a season ahead. We test the dependence of SST predictive skill in initialised hindcasts on the phase of AMOC at 26°N, invoking a seesaw mechanism driven by AMOC fluctuations, with positive SSTA North of 26°N and negative SSTA South of 26°N after strong AMOC and vice versa. We use initialised simulations with the MPI-ESM-MR seasonal prediction system. First, we use an assimilation experiment between 1979-2014 to confirm that the AMOC leads a SSTA dipole pattern in the tropical and subtropical North Atlantic, with strongest AMOC fingerprints after 2-4 months. Going beyond previous studies, we find that the AMOC fingerprint has a seasonal dependence, and is sensitive to the length of the observational window used, i.e. stronger over the last decade than for the entire time series back to 1979. We then use a set of ensemble hindcast simulations with 30 members, starting each February, May, August and November between 1982 and 2014. We compare the changes in skill between composites based on the AMOC phase a month prior to each start date to simulations without considering the AMOC phase, and find subtle influence of the AMOC mechanism on seasonal SST prediction skill. We find higher subtropical SST hindcast skill at 2-4 months lead time for JJA SSTA composites based on the AMOC phase at May start dates than for the full time period. In other regions and seasons, we find negligible impact of the AMOC seesaw mechanism on seasonal SST predictions due to atmospheric influence, calling for caution when considering such a mechanism. Our method shows that, for May start dates following strong AMOC phases, summer SST hindcast skill over the subtropics increases significantly compared to weak AMOC phases. This suggests that in the assessment of SST skill for a season ahead an eye should be kept on the initial AMOC state.

# 1 Introduction

Sea surface temperature (SST) variability at seasonal timescales has a significant impact on the weather and climate (Stockdale et al., 2011; Sutton and Hodson, 2005). Seasonal SST anomalies (SSTAs) in the tropics have been linked to the intensity and genesis of tropical cyclones and heatwaves (Coumou and Rahmstorf, 2012; Duchez et al., 2016b; Arora and Dash, 2016), and to fluctuations of marine resources (Stock et al., 2015); all of which have potentially important socio-economic consequences. Nevertheless, the mechanisms governing the predictability of seasonal SST changes are not well understood (Stocker, 2014). Correspondingly, seasonal predictions of SSTAs often show low skill, particularly over the extratropics (e.g. Arribas et al. (2011)).

Air-sea heat fluxes (ASFs) and Ekman-induced oceanic heat transport are important drivers of seasonal variability for SSTs (Bjerknes, 1964; Gulev et al., 2013). The North Atlantic Oscillation (NAO) is recognised as the main mode of climate variability at seasonal to interannual timescales in the North Atlantic (Deser et al., 2010), and a SST anomaly tripole with negatively correlated SSTA in the subpolar and tropical North Atlantic and positively correlated SSTA in between is seen as its major imprint on the ocean surface (Marshall et al., 2001). Part of the North Atlantic seasonal SST variability has also been attributed to the AMOC (e.g. Bryden et al. (2014); Zhang et al. (2019)). The AMOC is estimated to transfer about 1.3 PW ($10^{15}$ W) of heat northwards at 26°N (Johns et al., 2011). This heat transport, however, shows little meridional coherence at seasonal to interannual timescales (Bingham et al., 2007; Hirschi et al., 2007). Through local convergence or divergence of ocean heat transport (OHT, e.g. Cunningham et al. (2013); Borchert et al. (2018)), AMOC fluctuations could therefore influence the seasonal to interannual predictability of SST. The SST response to AMOC results in recurring large-scale patterns, generally known as AMOC fingerprints (Zhang, 2008).

Here, we examine the seesaw mechanism proposed by Duchez et al. (2016a) (henceforth D16), which links variations in strength of the Atlantic Meridional Overturning Circulation (AMOC) at 26°N and North Atlantic SSTs on monthly time scales. D16 analysed the relationship between AMOC observations at 26°N (Smeed et al., 2014) and ERA-Interim SST (Dee et al., 2011) during 2004-2014, finding a strong SST dipole pattern centred at 26°N following AMOC anomalies at 3-5 months lag. D16 proposed a dipolar response of SSTs to AMOC variability, in which a stronger than average AMOC at 26°N advects more heat northward, leading to colder waters in the tropics and warmer waters in the subtropics. Conversely, a weaker AMOC advects less heat northward of 26°N, building up heat south of 26°N, and leading to colder waters to the north and warmer to the south of 26°N. Hence, AMOC variations at 26°N were suggested as a precursor to SSTAs in the tropical and subtropical North Atlantic, implying a potential application on seasonal forecast systems. We evaluate to what extent the seasonal SST predictive skill in the North Atlantic is sensitive to the phase of AMOC before the prediction is made.

Recent studies have found improved hindcast skill in the North Atlantic region after considering known physical mechanisms into their seasonal prediction analysis. One way to incorporate physical mechanisms into prediction studies was by identifying and explaining times of low and high skill, including precursors of high skill, so-called *windows of opportunity* (Borchert et al., 2018; Mariotti et al., 2020). The present study focuses on oceanic processes that are arguably less noisy than atmospheric dynamics (Gulev et al., 2013), invoking an analysis of windows of opportunity.

Analysing an ensemble of yearly initialised hindcasts with MPI-ESM-LR covering 1901-2010, (Borchert et al., 2018, 2019) showed that the AMOC at 50°N influences the SST variability and predictability for several years, with higher skill after years of strong AMOC and vice versa. Borchert et al. (2018) perform a predictive skill analysis of SST conditioned to strong and weak OHT anomalies at 50°N separately, showing a robust influence of the ocean on windows of opportunity for decadal subpolar North Atlantic SST predictions. A similar analysis has not yet been performed on the seasonal time scale. Studies suggest, however, an influence of the ocean on seasonal predictability as well. In particular, seasonal SST potential predictability, i.e. the fraction of long-term variability that may be distinguished from the internally generated natural variability, was shown to improve for better represented ocean initial states in the tropical Pacific boreal winter (Alessandri et al., 2010), and in parts of the Atlantic (Balmaseda et al., 2013). We therefore pursue the establishment and explanation of windows of opportunity for seasonal predictions of North Atlantic SST, invoking the seesaw mechanism proposed by D16.

We apply a similar technique as Borchert et al. (2018) to evaluate the impact of the strength of the AMOC at 26°N on seasonal prediction of SST. Using simulations from the MPI-ESM-MR-based seasonal prediction system (Dobrynin et al., 2018) and invoking the seesaw mechanism proposed by D16, we examine whether predictions initialised following an anomalously strong AMOC event at 26°N are prone to show higher SST predictive skill north of this section. Likewise, predictions initialised after anomalously weak AMOC events at 26°N could show higher SST skill over the tropical region, to the south of this section, due to a local convergence of oceanic heat. Going beyond D16, we consider the seasonality of this connection. Knowledge about connections between SST prediction skill and preceding AMOC strength could be used by decision makers to narrow down the credibility of actual forecasts of North Atlantic SST (Borchert et al., 2019).

The paper is structured as followed: Section 2 describes the datasets and methods used in this paper. We verify the modelled AMOC against RAPID observations in Sect. 3.1. In Sect.3.2 we assess the influence of AMOC strength on seasonal SSTAs considering two different periods, and evaluate the contribution of seasonality and atmospheric processes. We carry out a predictive skill analysis in Sect.3.3, and assess the impact of considering the AMOC strength at the beginning of the prediction. Section 4 provides the discussion, followed by the summary and conclusions in Sect. 5.

# 2 Model and methods

## 2.1 Model description and the prediction system

We use retrospective seasonal predictions (hindcasts) with the coupled climate model MPI-ESM, in its mixed resolution (MR) setup (Baehr et al., 2015; Dobrynin et al., 2018) in the version as used for the CMIP5 simulations (Giorgetta et al., 2013). The oceanic component is the MPIOM ocean general circulation model, formulated on a tripolar grid with poles over North America, Siberia and Antarctica, with a nominal horizontal resolution of 0.4 degrees and 40 unevenly spaced vertical levels (Marsland et al., 2003; Jungclaus et al., 2013). The atmospheric component ECHAM6 runs at T63 horizontal resolution, i.e. approximate horizontal resolution of 200 km with 95 vertical levels, resolving the troposphere and the stratosphere up to 0.01 hPa (Stevens et al., 2013). Ocean and atmosphere are coupled daily without flux adjustments.

Initial conditions of the hindcasts are taken from a fully-coupled assimilation experiment with MPI-ESM-MR. In the assimilation experiment, full temperature and salinity fields in the ocean component were nudged (Dobrynin et al., 2018) towards the ORA-S4 reanalysis (Balmaseda et al., 2013). Temperature, vorticity, divergence, and surface pressure in the atmosphere component were nudged towards ERA-Interim (Dee et al., 2011), and the sea ice component was nudged to NSIDC observations (Comiso, 1995).

We use a 30-member hindcast ensemble initialised every February (FEB), May (MAY), August (AUG) and November (NOV) between 1982 and 2014 from the assimilation experiment (Dobrynin et al., 2018). We end our analysis in 2014, in order to compare to D16 using observations. After each initialisation, the ensemble members run freely for 6 months. The 30-member hindcast ensemble was generated by slightly modified initial conditions, using bred vectors in the ocean component with a vertically varying norm that allows for a full depth perturbation of the ocean (Baehr and Piontek, 2014). In the atmosphere, the diffusion coefficient in the uppermost layer is slightly disturbed to generate the ensemble.

## 2.2 Data Pre-processing and Statistical Methods

To evaluate the long-term SST dipole pattern dependence on AMOC variability, we use the assimilation experiment covering the period of January 1979 to December 2014. We choose the assimilation experiment over observations because of the short observational record of AMOC from the RAPID/MOCHA array that is available only from April 2004 (Cunningham et al., 2007). Our method therefore allows to constrain the seasonal cycle more robustly. Comparing our results back to the short observational record allows for an assessment of how model-based and observational dynamics compare. In the model, the meridional overturning transport is directly calculated using the 3-dimensional velocity field averaged at each latitude, and the AMOC is defined as the vertical maximum of the stream function. We verify the modelled AMOC using observations from the RAPID array at $26°N$. The RAPID AMOC is defined as the sum of three components: the Florida Strait transport, the surface Ekman transport (EKM), and the geostrophic upper-mid-ocean transport. A detailed description of the calculation of the individual components is provided in Smeed et al. (2018).

We evaluate the atmospheric contribution to the SST variability using the Ekman transport (EKM) and air-sea heat fluxes. We evaluate both the EKM relationship to SST, as well as the AMOC without its EKM component, i.e. AMOC-EKM (Mielke et al., 2013). EKM is calculated from the zonal wind stress component $\tau_x$ integrated over the Atlantic, i.e. $EKM = -\int \frac{\tau_x}{\rho f} dx$, where $\rho$ is the reference density ($1025\,\text{kg m}^{-3}$) in MPIOM and $f$ is the Coriolis parameter. For ASF we use the total surface heat fluxes over the ocean, which include shortwave, longwave, latent and sensible heat fluxes. ASF is parameterized as described in Marsland et al. (2003), with fluxes defined positive downward.

To further analyse the influence of AMOC on SST variations, we calculate the convergence of OHT with respect to two latitude bands encompassing a tropical ($10.5°$ - $22.5°N$), and a subtropical ($28.5°$ - $40.5°N$) region. These latitude bands are the same used to define Box 1 and Box 2 in Fig. 4.a. Following Jayne and Marotzke (2001), we calculate the OHT as the zonal and vertical integral of the heat flux across an east-west section through the Atlantic Basin, i.e.

|  | AMOC | EKM | AMOC-EKM |
|---|---|---|---|
| MPI-ESM-MR | $18.42 \pm 2.55$ (2.79) | $3.08 \pm 1.61$ (3.36) | $15.34 \pm 2.28$ (2.55) |
| RAPID | $17.02 \pm 2.95$ (3.90) | $3.56 \pm 1.46$ (2.26) | $13.43 \pm 0.96$ (2.42) |

**Table 1.** Transports mean values, standard deviations and seasonal ranges (in parentheses) for the model (1979-2014) and observed AMOC (2004-2014). All values in Sv.

$$OHT = \rho_o c_p \int\limits_{x_W}^{x_E} \int\limits_{-H(x,y)}^{0} v(x,y,z)\theta(x,y,z)\,dz\,dx$$

where $\rho_o$ is a reference density, $c_p$ the specific heat capacity of sea water, H is the water depth, x stands for longitude and y for latitude, z is the water column depth, $x_E$ and $x_W$ are the eastern and western limits of the section, v is the meridional velocity, and $\theta$ is the potential temperature in degrees Celsius.

We calculate monthly means of AMOC, EKM, SSTA and air-sea heat fluxes. Our main data set consists of model output, in addition to AMOC observations from RAPID and 'observed' SST from the ERA-Interim reanalysis (Dee et al., 2011). This data set is deseasoned by removing the 12-month climatology obtained from the monthly data, and the linear trend is removed as an idealised approach to remove the externally forced signal from the time series and focus on internal variability. We refer to these detrended, deseasoned quantities as anomalies. Seasonal means are defined as December-January-February (DJF) for winter, March-April-May (MAM) for spring, June-July-August (JJA) for summer and September-October-November (SON) for autumn.

To assess the variability of the AMOC fingerprint and to evaluate its role on seasonal SST predictability, we perform lagged correlations from 0 up to 12 months, with the AMOC leading SSTA. Additionally, we compute lagged correlations for ASF, EKM and AMOC-EKM leading SSTA to explore the relative contributions of atmospheric and oceanic dynamics to SSTA changes. For our hindcast skill analysis, we assess predictive skill of the hindcast simulations against the ERA-Interim data with the point-wise Anomaly Correlation Coefficient (ACC, Collins (2002)). We calculate statistical significance of our findings using a Monte-Carlo bootstrapping method. The process consists of 1000 bootstraps with replacement on the time-dimension at the 95% confidence level.

## 3   Results

### 3.1   Verification of the AMOC in the assimilation experiment

We evaluate the AMOC seasonal cycle using both anomalies and full values. To show the spread of the annual climatology, grey lines in Fig. 1.a, c, e represent anomalies w.r.t. the mean transport of a given year calculated for the full time series (1979-2014), and smoothed with a 3-month running average to highlight the seasonal cycle. The observed AMOC shows minimum transport in March and maximum in August (Fig. 1.a, b). Minimum transport for the modelled AMOC is achieved slightly later,

in April-May, while it peaks twice in August and December. The seasonal cycle for both the observed and the modelled AMOC agree with the ones discussed by Mielke et al. (2013) using RAPID data from 2004-2010 (Cunningham et al., 2007) and a high resolution MPI ocean model spanning the same period. For EKM (Fig. 1.c, d), the seasonal cycle for observations and model are slightly out of phase, but both show a clear maximum in summer (July-August) and minimum in spring (March-April). The seasonal range for the modelled EKM is 3.36 Sv, compared to 2.26 Sv for the observations. The opposite is found for the AMOC seasonal range, which is smaller for model (2.79 Sv against 3.90 Sv, Table 1 1). These differences in range and phase for AMOC and EKM can explain the seasonal cycle of AMOC-EKM, with minimum in July and maximum in November (Fig. 1.e, f). Time series of observed and modelled AMOC, EKM and consequently AMOC-EKM are in reasonable agreement with a correlation of 0.67 and 0.66, respectively (Fig. 1.g). There is no relevant effect of the mean state on these findings, which is why we use anomalies from now on.

## 3.2 Impact of AMOC fingerprints on North Atlantic SST variability

### 3.2.1 The RAPID decade

Here, we compare the observed AMOC fingerprints discussed in D16 with those found in the assimilation experiment for the RAPID period April 2004 to March 2014 (c.f. D16's Fig.3). We calculate lagged correlations up to 12 months, with the AMOC leading (Fig. 2 for maximum -month lag).

We find that during the RAPID decade a dipole pattern represents the influence of AMOC on Atlantic SST variability in the assimilation experiment up to 7 months ahead with positive correlation in the subtropical and negative correlation in the tropical regions, similar to D16. Specifically, this pattern is composed of a large zonal band of anticorrelation located between 5 and 26°N, from the African coast towards the Gulf of Mexico, and a smaller positive correlation lobe between 26 and 40°N (Fig. 2). This dominant SSTA correlation pattern evolves over time. Lags 0 and 2 show maximum positive correlations of the order of 0.6 mostly in the western side of subtropical lobe near the US coast, as opposed to maximum negative correlations of similar magnitude mainly at the eastern side of the tropical North Atlantic, close to northwestern Africa. The magnitude of the correlation (anticorrelation) drops to maximum of 0.4 (minimum of -0.5) with increase in lag. With increasing time lag (5-7 months specifically), the subtropical lobe of positive correlation shows a displacement towards the east.

The correlation pattern for the subpolar region is also pronounced, however the strongest negative correlations of -0.4 are only present up to 2 months lag (Fig. 2.a, b). These negative correlations have been previously associated with the NAO imprint in the Atlantic (Fan and Schneider, 2012; Oelsmann et al., 2020), and are not explained by D16's physical mechanism which we investigate in this study. D16's physical mechanism attributes an active role of ocean heat advection on the SST variability at the timescale of a few months, due to anomalous convergence or divergence of OHT. Therefore, we restrict our analysis to the AMOC influence on SST over tropical and subtropical North Atlantic, and exclude the subpolar pattern from our analysis.

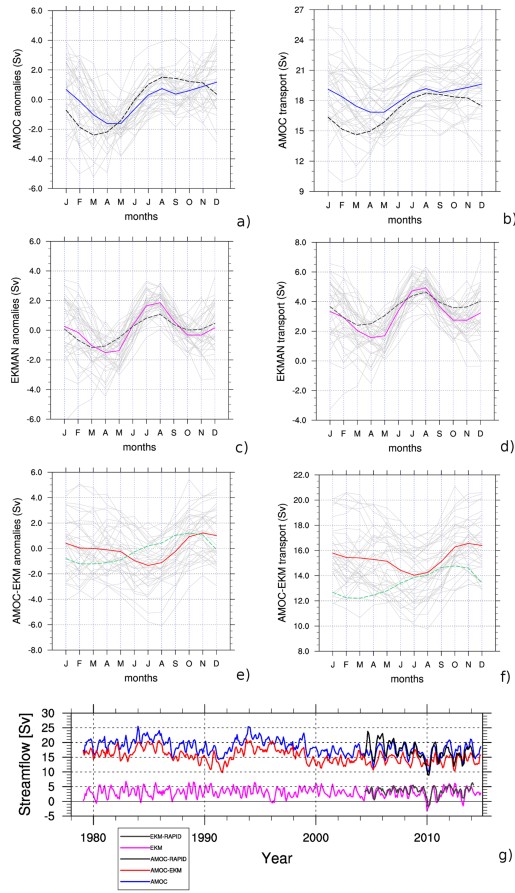

**Figure 1.** The AMOC in the assimilation experiment. Climatology of the maximum AMOC transport at 26°N in the assimilation experiment, smoothed with a 3-month running mean and the annual cycle removed (spanning 1979-2014), for anomalies (a, c, e), and full values (b, d, f) as labelled. The highlighted full coloured lines represent the mean transport values, whereas each light grey line represents a given year. The dashed lines correspond to the mean value of observed AMOC. g) modelled AMOC at 26°N (blue line), AMOC-EKM (red line) and EKM (magenta line); the observed AMOC (black line) and EKM as the component in the RAPID data (grey line).

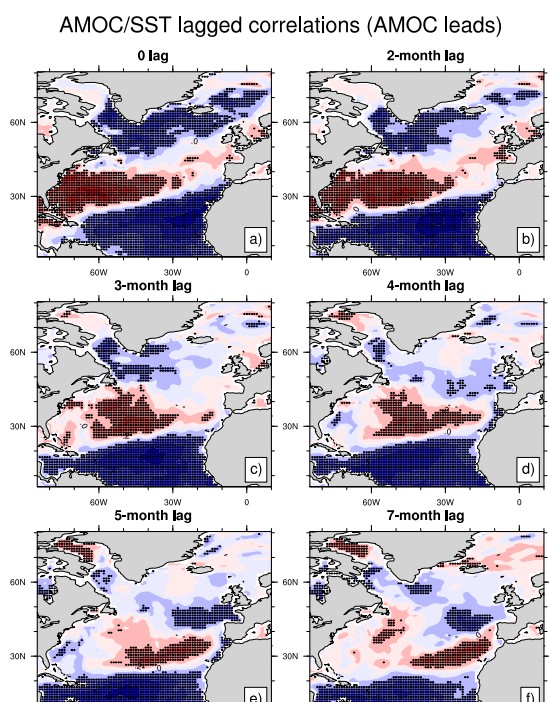

**Figure 2.** Lagged correlations between AMOC at 26°N and North Atlantic SSTA during the RAPID period (2004-2014), with the AMOC leading (a-f, as labelled) in the assimilation experiment. The stippling represents significant correlations at the 95% confidence level, calculated from 1,000 bootstrap samples.

### 3.2.2 Investigating a 30-year period

We now analyse the impact of AMOC on SSTs over the full 36 year period for which the assimilation experiment is available, to assess the consistency of the previous results (Fig. 2) over a longer period. The SST dipole pattern for the full 36-years and RAPID period (Fig. 3) hold mostly similar spatio-temporal characteristics. The longer period shows, however, lower correlation values than during the RAPID period, particularly over the subtropics and at long lead times >3 months.

The subtropical lobe shows a consistent positive correlation throughout lags 1 to 7 months, with higher correlation at lag 0 (Fig. 3). At lags >3 months, however, these correlation values become statistically insignificant. The tropical lobe of the dipole shows minimum negative correlations ranging from -0.29 and -0.37, which are comparable with the magnitude observed during the RAPID period for the same region. The SST dipole also shows a dependency on the time period that is analysed, which is in agreement with findings presented by Alexander-Turner et al. (2018). Analysing this time-dependency further by computing running lagged-correlations for 10- and 15-year windows for lags 1 and 3 months (see video supplement), we find that the 1990s

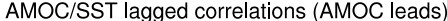

AMOC/SST lagged correlations (AMOC leads)

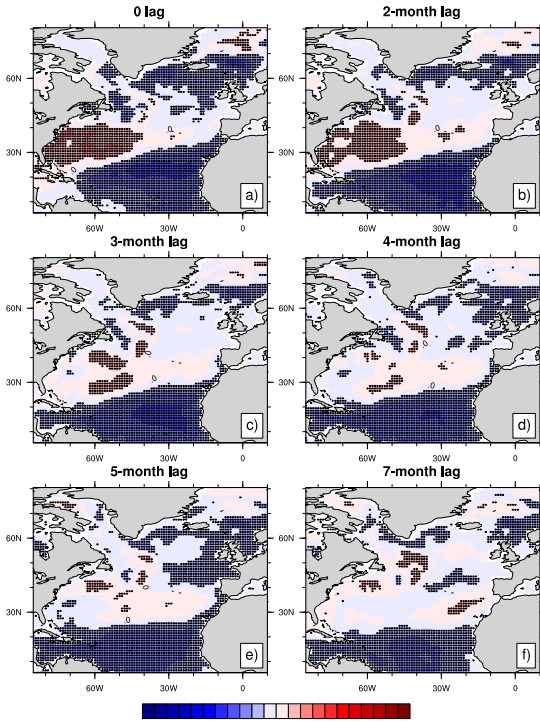

**Figure 3.** Lagged correlations between AMOC at 26°N and North Atlantic SSTA during 1979-2014, with the AMOC leading (a-f, as labelled) in the assimilation experiment. The stippling represents significant correlations at the 95% confidence level, calculated from 1,000 bootstrap samples.

tend to show a less clear AMOC-SST correlation pattern in comparison to both the 80s and the RAPID period, particularly for the subtropical lobe of the SST dipole.

To further explore the sensitivity of AMOC fingerprints to the length of the observational window used, we show the AMOC-SST relationship averaged over two regions comprising the dipole lobes (Fig. 4). We define those as tropical lobe: Box 1 (10.5°
185   - 22.5°N, 22° - 55°W), and subtropical lobe: Box 2 (28.5° - 40.5°N, 40° - 70°W, cf. Fig. 4a). We focus on positive lags, which represent the AMOC-influenced SST fingerprints. Elaborating on findings based on spatial fields (Figs.2,3), we here show spatially aggregated SST variability (Fig. 4). As before, AMOC fingerprints over the RAPID period are stronger than over the full time series. We find high anticorrelations up to 5-month lag over Box 1, ranging from -0.57 at 5-month to maximum magnitude of -0.69 at 2-month lag. In contrast, when the full time series is considered, values drop to the order of -0.4. This
results in a significant skill difference between the RAPID and the full period for lags 1-5 months, evaluated by non-overlapping uncertainty envelops for the two correlation estimates (Fig. 4b). Similarly, we find high correlation values above 0.6 up to 2-month lag over the RAPID period for Box 2, which drop to 0.24 at 5-month lag. The magnitude of correlations for Box 2 over

the full time series reaches a maximum of 0.25. Correlation estimates for box 2 are significantly different between the two periods for lags 0-4 months (Fig. 4b). Weakened AMOC fingerprints with reduced correlation of AMOC to SST, particularly during the 90s (as exemplified in the time series for AMOC and SST; Fig. 4c, d), are likely responsible for the decline of the correlations computed for the full time series.

D16's physical mechanism suggests that via convergence (divergence) of OHT in the subtropics (tropics), a strong AMOC at 26°N drives the subtropical and tropical SST variability at a maximum of 2-5 months lead time. To test the physics behind this hypothesis, we assess the convergence of OHT for the latitudinal bands corresponding to Box 1 and 2 (Fig. 4.e, f, respectively). We define OHT convergence as $\delta OHT = OHT_{SouthernBoundary} - OHT_{NorthernBoundary}$, i.e. heat flow into the latitudinal band minus heat flow out of the band (as in Borchert et al., 2018). This analysis shows significant negative correlation of AMOC with OHT convergence in the latitudes of the tropical lobe (Fig. 4e), showing that AMOC-related outflow of heat represents a relevant driver of heat convergence changes in the area. Further, AMOC is strongly positively correlated to OHT convergence in the latitudes of the subtropical lobe of the AMOC fingerprint, indicating a substantial impact of AMOC-related heat transport on oceanic heat convergence there. AMOC explains more of the heat convergence in the subtropical box 2 than in the tropical box 1, indicating a larger value of knowledge of AMOC for heat convergence and consequently SST in box 2 than in box 1. While a detailed heat budget for the exact boxes studied here would certainly be interesting and add more detail to this assessment, its calculation is complex and beyond the scope of this work. For now, OHT convergence analysis indicates an influence of AMOC on oceanic heat accumulation north and (less so) south of the AMOC latitude, but also illustrates that other factors contribute to accumulation of heat in the ocean that overturning does not account for. Such factors could be direct heat fluxes from the atmosphere, zonal or vertical heat fluxes in the ocean, or meridional influences that operate on longer lags than examined here. For example, an equivalent analysis to Figure 4 e and f using SST in the $\delta OHT$ latitudinal bands instead of box 1 and 2 (not shown) shows a correlation decrease for both bands compared to the use of box 1 and 2, indicating a non-negligible contribution of zonal heat transport in both cases. Since correlation is significant for box 2 in both cases, though, we maintain our conclusion that AMOC-related meridional ocean heat convergence contributes significantly to SST changes there. We will therefore continue to study the AMOC as a predictor of SST, with the accompanying limitations pointed out above.

### 3.2.3   The seasonal dependence

Going beyond previous work (Duchez et al., 2016a; Alexander-Turner et al., 2018) in investigating the variability of the SST dipole pattern, we analyse the role of SST seasonality. Using the assimilation experiment for the period of 1982-2014, we perform correlations of the AMOC anomalies at a given month with the mean seasonal SSTA 2-4 months ahead (Fig. 5). By doing so, we provide a detailed view of the temporal variability of the SST dipole pattern, enabling an assessment of the contribution of drivers other than AMOC that could potentially affect the SST variability to the observed pattern.

We find a strong fingerprint in spring (MAM), with average (maximum) correlation of the order of 0.4 (0.52) (Fig. 5). During summer (JJA), the fingerprint is less pronounced than in spring, with lower average (maximum) correlation magnitudes of around 0.3 (0.44), but still clearly identifiable. In contrast, we find that autumn and winter seasons lack a characteristic

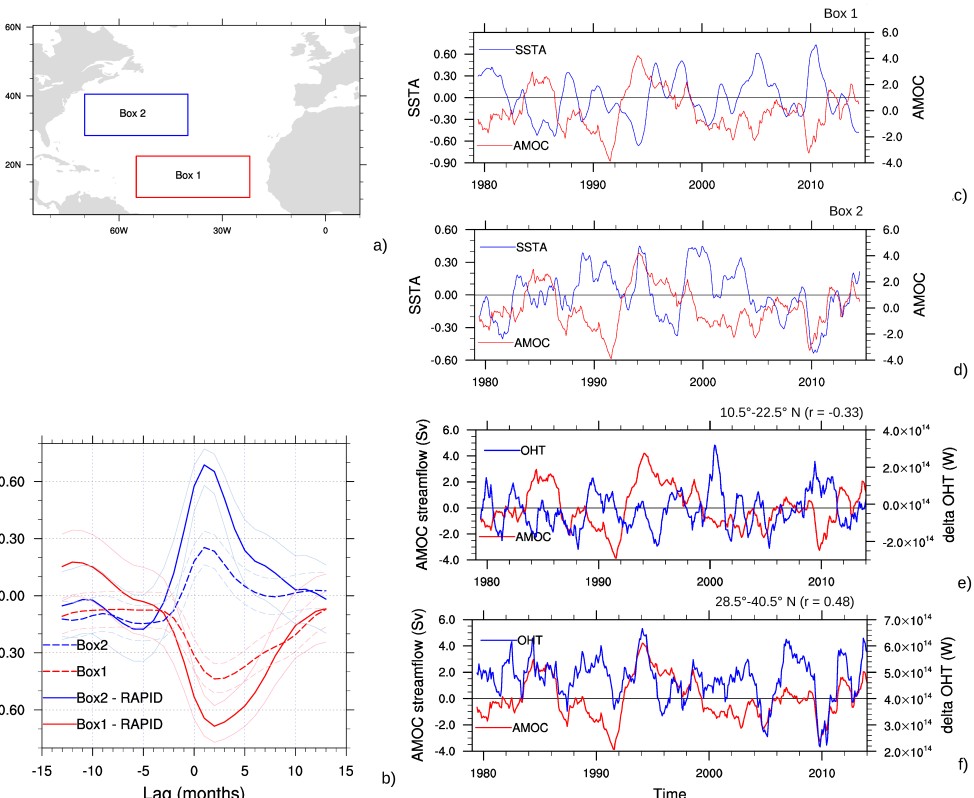

**Figure 4.** Relationship of AMOC at 26°N, SSTA and $\delta$ OHT over two regions in the North Atlantic in the assimilation experiment: a) regions used for averaging SST; b) lagged correlations between AMOC and SSTA over Box 1 (red lines) and Box 2 (blue lines). Bold lines correspond to the RAPID period, dashed lines to 1979-2014, and light colours show the significance at 95%; c) time series for AMOC and SSTA averaged for Box 1 and d) for Box 2, both smoothed with a 12-month running mean; time series for AMOC and $\delta$ OHT w.r.t. e) 10.5° - 22.5°N, and f) 28.5° - 40.5°N.

dipole pattern (Fig. 5.c, d), showing instead only a narrow region of negative correlations over the subtropics of the order of -0.2 (-0.1) for winter (autumn). The absence of a dipole pattern in autumn and winter may suggest the influence of atmospheric drivers that could potentially supersede the AMOC fingerprints during these seasons. Moreover, we find similar characteristics

for the AMOC fingerprints using the ensemble mean hindcasts (not shown), which is especially relevant for interpreting the SST predictive skill analysis in section 3.3.

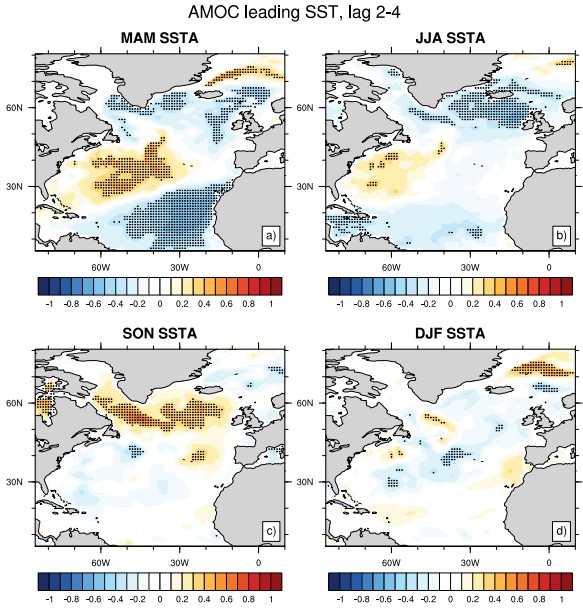

**Figure 5.** 2-4 month lagged correlations between the SSTA over the North Atlantic and the AMOC at 26°N during 1982-2014, with the AMOC leading (a-d, as labelled) in the assimilation experiment. For example, in a) AMOC in January is correlated to spring SSTAs. The stippling represents significant correlations at the 95% confidence level, calculated from 1,000 bootstrap samples.

### 3.2.4 The atmospheric contribution

At the seasonal timescale, much of the SST variability in the North Atlantic is the response to atmospheric forcing (Deser et al., 2010). The two main processes responsible for the atmospheric imprint in the large-scale SST variability are anomalous ASFs

and EKM-induced oceanic heat transport. The former is known to induce the tripolar SST pattern (Fan and Schneider, 2012), and is mostly forced by the NAO (Cayan, 1992; Marshall et al., 2001). Anomalous EKM may also contribute to SST variability, especially over regions of strong temperature gradients such as the Gulf Stream (Deser et al., 2010). Fig. 5 shows that AMOC fingerprints have a seasonal dependence. One possible explanation for this seasonality is a stronger atmospheric role on the SST variability in comparison to the AMOC influence, depending on the season. To further explore these interactions, we

assess the relative contributions of ASFs and EKM to the SST variability.

We compute correlations between cumulative ASF anomalies and SSTA for 2 months (where ASF leads) for each seasonal mean SSTA (Fig. 6), thus highlighting regions and seasons where the atmosphere strongly contributes to SST variability. ASFs are defined as positive into the ocean, i.e. positive correlations with SST are interpreted as SST response to atmospheric heat fluxes, and vice versa. Consequently, significant positive correlations between cumulative ASF and SSTA indicate significant atmospheric contribution to SST changes that, should they overlap with AMOC fingerprints identified in Figure 5, indicate a role for the atmosphere in these regions that are potentially unpredictable. As such, this analysis forms an important step towards the assessment of seasonal SST predictions.

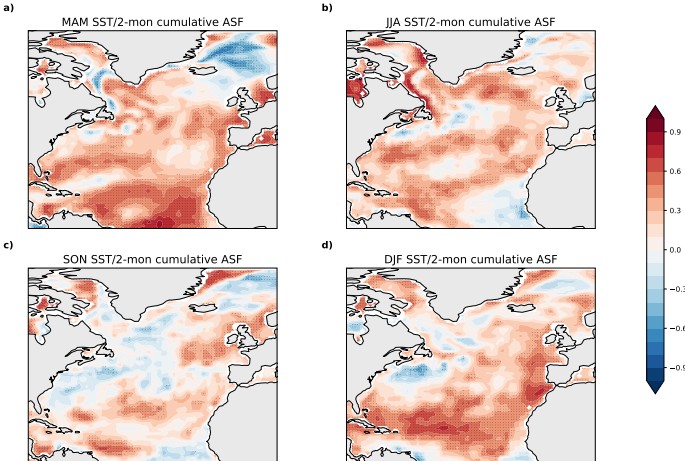

**Figure 6.** Correlations between the 2-month cumulative ASFs and SSTAs in spring (a), summer (b), autumn (c) and winter (d) in the assimilation experiment. As an example, in a) January and February ASF anomalies are correlated to MAM SSTA. The stippling represents significant correlations at the 95% confidence level, calculated from 1,000 bootstrap samples.

We find overall positive correlations between SSTs and ASFs (i.e. atmosphere forcing to the ocean) on the seasonal time scale (Fig. 6), with a few exceptions, e.g. over the Gulf Stream region. We compare these results to Fig. 5, to evaluate whether regions of positive strong ASF-SST correlations coincide with those of AMOC fingerprints. During spring and winter, we find positive and significant ASF-SST correlation located over the tropical lobe of the AMOC fingerprint, while correlations found on the subtropical lobe are mostly not significant or negative (i.e. ocean forcing the atmosphere). In summer and autumn, we find negative or weak positive correlations on larger parts of the tropical and subtropical lobes, indicating potential influence of AMOC over ASFs during these seasons.

In addition to ASFs, Ekman transport is an important contributor to short-term SST variability (Frankignoul, 1985). EKM is the wind-driven component of the overturning in the ocean, forming the full AMOC signal together with the overturning in the ocean interior, to which usually most of the northward heat transport is attributed (Ferrari and Ferreira, 2011). For spring SSTAs, we find a strong contribution of EKM to the AMOC fingerprint, illustrated by EKM-SST 2-4 month lagged correlations holding a well-define tripole (Fig. 7e), in agreement with D16. For summer SSTAs, however, EKM weakly contributes to the

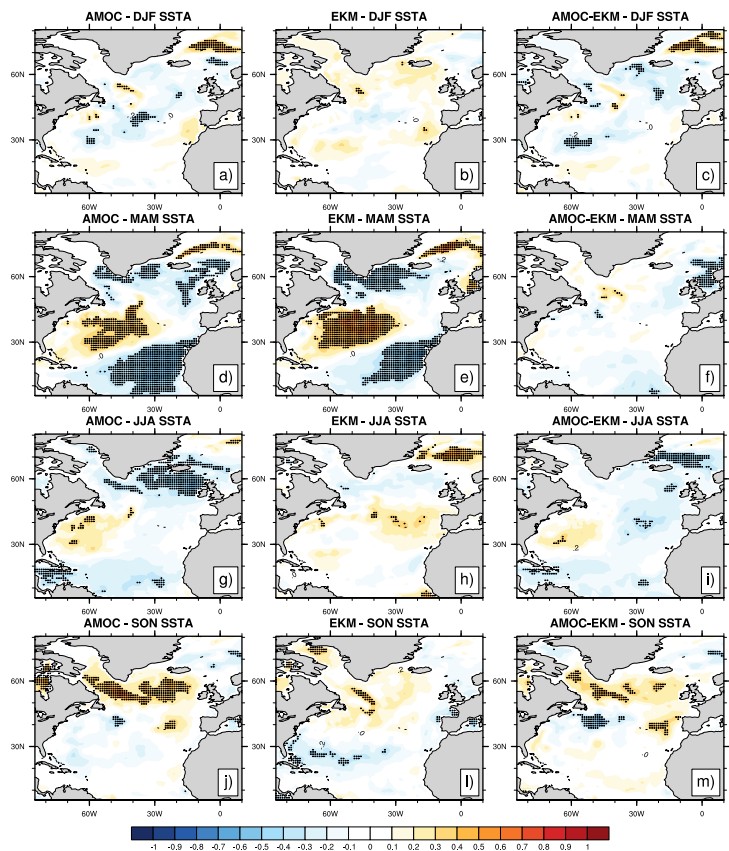

**Figure 7.** Correlations between SST seasonal means and AMOC (a, d, g, j), EKM (b, e, h, l) and AMOC-EKM (c, f, i, m) at 2-4 month lag (with the transport leading SST) covering the 1979-2014 period in the assimilation experiment. The stippling represents significant correlations at the 95% confidence level, calculated from 1,000 bootstrap samples.

subtropical lobe of the dipole – this contribution is mostly not significant (Fig. 7h). In the other seasons, SST variability in the tropics and subtropics seems to be less influenced by EKM, as shown by the weak correlation pattern for EKM (Figs.7b,l) as well as the high similarity in the SST patterns for AMOC and AMOC-EKM (Figs.7a, c, j, m).

In summary, we present here in further detail the implications of the AMOC fingerprint on North Atlantic SSTs by assessing the atmospheric contribution in terms of cumulative ASFs and EKM on the seasonal SST variability. At time lags where strong AMOC fingerprints occur (2-4 months), we find significant contribution from the atmosphere to SSTA in many regions during winter (ASFs) and spring (ASFs and EKM). Since the AMOC fingerprints are generally weak in autumn (Fig. 5), implying a small overall influence of AMOC on SST during that season, this leaves us with a AMOC impact on seasonal SSTA in the region of the AMOC fingerprint during boreal summer. These findings might have implications for seasonal predictions of tropical and subtropical SST.

## 3.3 Seasonal hindcast skill

Based on the AMOC fingerprint variability and sensitivity we assessed above, we now test whether considering the AMOC strength at 26°N at the beginning of the prediction may improve the SST predictive skill in the North Atlantic. We particularly focus on the start month dependence of the predictive skill by analysing 30-member hindcast ensembles started every February, May, August and November separately. The hindcasts initialised in FEB, MAY, AUG and NOV yield 2-4 months lead time SST targets MAM, JJA, SON and DJF, respectively. This allows us to build directly on the previously presented results.

All four seasons differ spatially in ACC skill of SSTs for 2-4 months lead time (Fig. 8a-d). Skill over the subtropics is lower than over the tropics for all start dates. SON SSTs show the lowest ACCs over the subtropics, while DJF and JJA exhibit mediocre, and MAM high subtropical SST skill. These results are robust across different lead times, showing similar spatial characteristics, albeit lower skill, for SST ACCs for 3-5 months lead time (not shown).

### 3.3.1 The role of the AMOC

We now assess the role of the AMOC fingerprints in the SST predictive skill with particular attention to strong and weak phases of the AMOC at 26°N, as dominant AMOC phases imply enhanced memory in the climate system through heat convergence and storage and thus elevated predictability, and vice versa (cf. Fig. 4e, f). To this end, we analyse SST hindcast skill at 2-4 months lead time for phases of strong and weak AMOC at 26°N separately (similar to Borchert et al. (2018)). Strong and weak AMOC phases are defined as stronger or weaker than average AMOC a month before the initialisation of the respective hindcast, however our results are not particularly sensitive to the exact choice of threshold for the definition of strong and weak AMOC phases (not shown). This analysis is performed for all start months separately. We therefore examine changes in the predictive skill for SST over the tropical and subtropical North Atlantic as modulated by the AMOC through the physical mechanism proposed in D16. If SST hindcast skill were to be influenced by the D16 mechanism, we would expect higher skill in the subtropical box 2 after strong AMOC phases and higher skill in the tropical box 1 after weak AMOC phases, due to heat accumulation in the respective box.

We find an overall subtle influence of AMOC on seasonal SST hindcast skill. The impact of the D16 mechanism on seasonal SST predictions differs between box 1 and 2, as well as heavily depends on the considered season. After strong AMOC phases, SST hindcast skill in the North Atlantic appears to be generally slightly increased compared to ACCs considering the full period (Fig. 8e-h), although there are some regions and seasons that show slightly higher than average skill after weak AMOC phases (Fig. 8i-l), such as the tropical North Atlantic during summer or winter.

The nuances of the impact of AMOC on seasonal SST hindcast skill are exemplified by skill difference plots between weak and strong AMOC phases (Fig. 8m-p). Summer SSTAs show a ACC difference between strong and weak AMOC phases that is in line with the expectation from the D16 mechanism (Fig. 8o), with higher skill in box 2 after strong AMOC and higher skill in box 1 after weak AMOC, and vice versa. All other seasons show either marginal impact of AMOC on SST skill (winter and autumn), or even a reversal of the mechanism with low subtropical skill after strong AMOC and vice versa (spring).

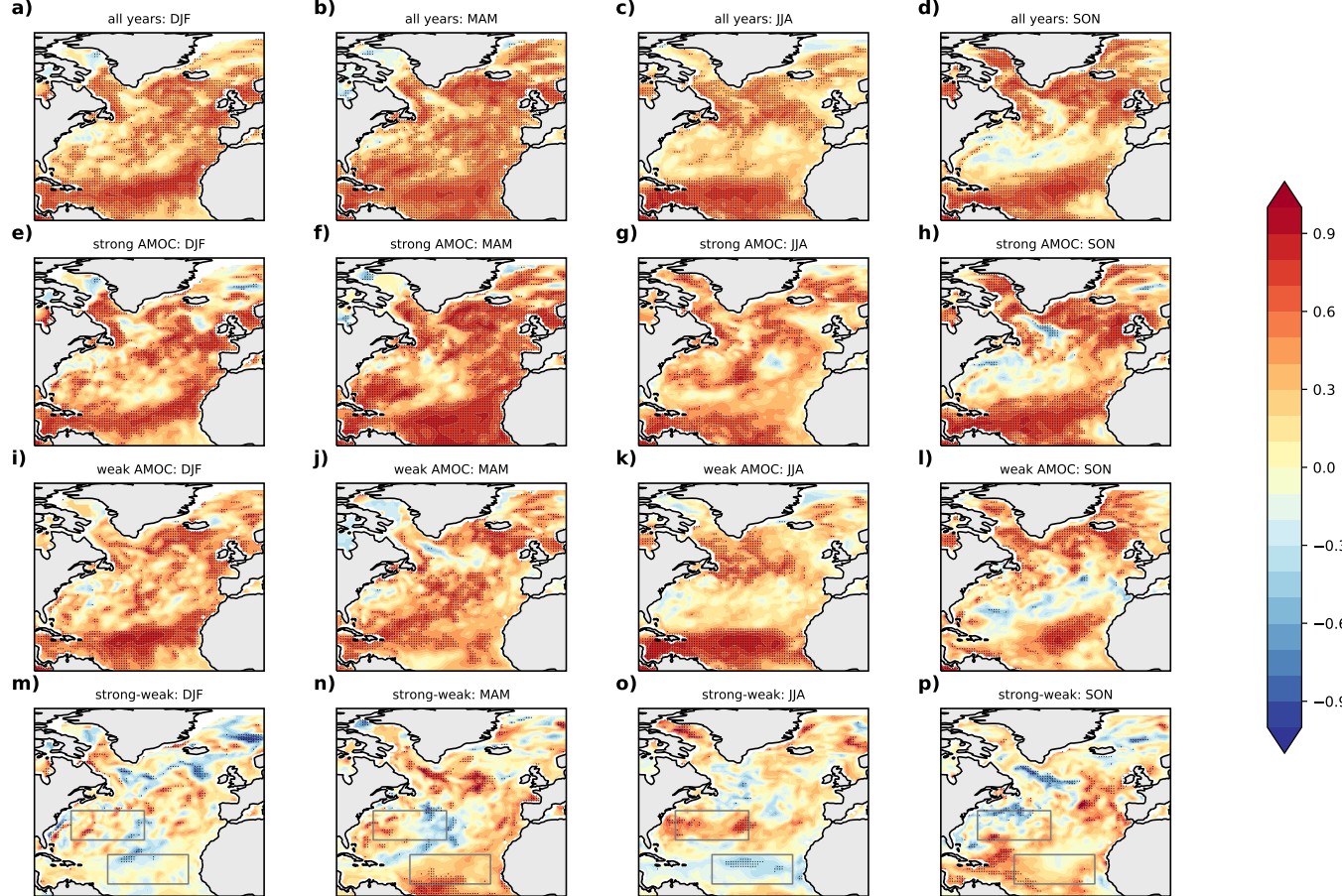

**Figure 8.** SST ACCs against ERA-Interim at 2-4 months lead time. Top row shows ACCs including the full 33-year time series (1982-2014) for NOV (a), FEB (b), MAY (c) and AUG (d). Next two rows show SST ACCs for composites based on either strong (e, f, g, h) or weak AMOC phases (i, j, k, l) 2-4 months before the SST mean, as labelled; e.g. DJF SST composites are based on the strength of AMOC at 26°N in October. Each column shows ACCs for a particular season, starting with winter (DJF) on the left. The bottom row shows the difference between ACCs in strong and weak phases, with boxes as used in Fig.4 to highlight regions where we expect the seesaw mechanism to influence the skill. For ACC, the stippling represents significant correlations at the 95% confidence level, calculated from 1,000 bootstrap samples. For the difference plots, significance is calculated at the 90% confidence level.

According to the D16 mechanism, high skill would be expected in box 1 after weak AMOC phases due to the accumulation of heat. Conversely, high skill would be expected in box 2 after strong AMOC due to accumulation of heat. As argued earlier in the manuscript, the strongest influence of AMOC on SST skill would be expected during JJA, because ASFs contribute significantly to SSTA during DJF and MAM, EKM drives much of MAM SSTA, and SON shows a generally weak connection of subtropical and tropical SSTAs to AMOC (Section 3.2). In addition, our analysis of ocean heat convergence indicated a stronger influence of meridional heat convergence in the ocean on subtropical box 2 than the tropics. We find (barely) significantly increased SSTA skill after weak AMOC compared to all years, strong AMOC, and persistence – defined by non-overlapping confidence intervals – only during summer (Fig. 9). Box 2, however, exhibits clearly and significantly elevated skill after strong AMOC phases during summer, beating all reference forecasts. This is a particularly striking finding since skill in box 2 is on average less significant than in box 1, and considering the D16 mechanism with the nuances presented in this study can help to elevate seasonal SST hindcast skill to significance. Thus, in combination, the expectations formulated earlier are met in our skill analysis: we find the strongest influence of AMOC on seasonal SST hindcast skill in the sense of the D16 mechanism during JJA in box 2 (Fig. 9).

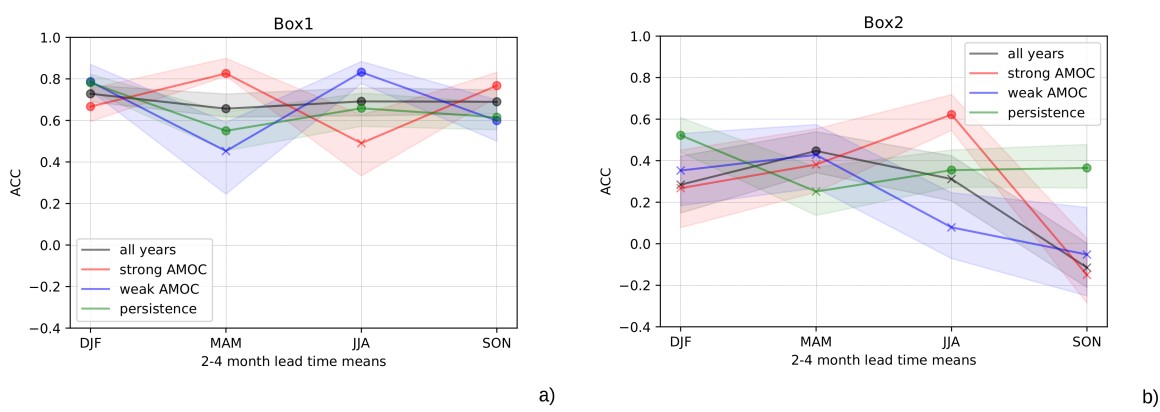

**Figure 9.** SST ACCs against ERA-Interim at 2-4 months lead time averaged over the regions shown in Fig. 4a. a) Box 1 (10.5° - 22.5°N, 22° - 55°W), and b) Box 2 (28.5° - 40.5°N, 40° - 70°W. Black lines represent the ACCs considering the full time series (1982-2014), red lines for strong, and blue lines for weak AMOC phases. A baseline persistence forecast can be seen in green. The shaded areas indicate the interquartile ranges and markers indicate significance at the 95% level (circles for significant, crosses for insignificant correlations).

## 4 Discussion

While a number of papers show evidence for robust AMOC fingerprints on North Atlantic SSTs at decadal and longer time scales (e.g. Zhang (2008); Muir and Fedorov (2015); Borchert et al. (2018)), only recently the extent to which AMOC influences

SST at seasonal time scales has been addressed (Duchez et al., 2016a; Alexander-Turner et al., 2018). In this study we explore the influence of AMOC strength at 26°N on North Atlantic SST seasonal variability and predictability in the MPI-ESM-MR model. We specifically test whether seasonal model AMOC fingerprints agree with the physical mechanism proposed in D16, and could therefore be considered in prediction analysis to condition seasonal SST hindcast skill over North Atlantic tropics and subtropics on the AMOC phase at the start of the prediction. Our findings suggest that the strength of AMOC is a potential regional source of subtropical SST predictability, most prominently during summer (JJA), by controlling the variability of heat advection north or south of 26°N. In other seasons and regions, the impact of AMOC on seasonal SST predictions is limited, for example by strong Ekman transport influence during spring (MAM) and dominant heat fluxes from the atmosphere during winter (DJF).

In line with D16, we find pronounced AMOC fingerprints at 2-5 months lag when considering the RAPID period (2004 - 2014). Going beyond this study, however, we find that AMOC fingerprints are sensitive to the length of the observational window used (as also noted by Alexander-Turner et al., 2018). Although our findings are in good agreement with D16 when we restrict the analysis to the most recent decade (cf. Fig. 2), we find less pronounced AMOC fingerprints with respect to the full time series back to 1979, at a maximum of 2-4 months lag. A possible reason for these differences could be multidecadal changes in AMOC variability and their imprint on SST (e.g. Ba et al. (2014); Knight et al. (2005); Borchert et al. (2018)). The RAPID period corresponds to a period of warm SST over the North Atlantic (Zhang, 2007, we find this in both model and observations), due to changes in AMOC dynamics at the multidecadal time scale, resulting in stronger OHT that could potentially enhance the AMOC influence at the seasonal timescale.

Alexander-Turner et al. (2018) found a similar time dependence using a 120-year long preindustrial control simulation with HadGEM3-GC2. They tested the robustness of the AMOC fingerprints on SST through time, finding good agreement with D16 at the 5-month lag, when taking the mean of 11-year segments of the full time series. However, when considering the full 120-year time series, this agreement was overall lower than when analysing the 11-year segments. Likewise, we find weaker AMOC fingerprints when analysing 30-year segments selected from the MPI-ESM-MR historical simulation (not shown). In tandem with the work from Alexander-Turner et al. (2018), our work therefore illustrates the importance of placing analyses on observed AMOC influence of SST in the broader temporal context using model simulations.

A key aspect that distinguishes our analysis from previous studies is that we find a significant seasonal dependence on the AMOC fingerprints. This dependence is coherent in both initialised and free-running model (not shown), with the strongest AMOC fingerprints occurring during spring and summer. In line with Alexander-Turner et al. (2018), we argue that a main driver for this seasonal dependence is the contribution of stochastic atmospheric variability, and in a lesser extent the Ekman transport. This has a direct implication on the consideration of the physical mechanism in our seasonal prediction system, which thus depends on the initialisation month.

The impact of this seasonal dependence can be illustrated as the distinguished effects of the physical mechanism on the hindcast skill for each start month (cf. Fig. 8). Our results suggest that summer (JJA) stands out as promising target season for the subtropics, given the relatively weak influence of EKM (strong in MAM) and ASFs (strong in all seasons, but with a "hole" in the subtropics during summer), which may decrease the influence of the seesaw mechanism on SSTAs for those

seasons. Such windows of opportunity for skilful summer SST predictions (Mariotti et al., 2020) in turn may benefit winter NAO predictions, with consequent influences on the storm track activity starting from October (e.g. Cassou et al. (2004)), as

well as on the development of Blocking regimes (e.g. Guemas et al. (2010)), and extreme events (e.g. Arora and Dash (2016)).

After weak AMOC phases, we find high tropical SST hindcast skill during DJF and JJA (among which the JJA improvement in line with the D16 mechanism is significant, cf. Fig. 9), in particular over the central hurricane main development region. These improved SST predictions over the hurricane main development region could be extremely beneficial for assessing seasonal hurricane formation probabilities (Saunders and Lea, 2008). We also find enhanced prediction skill in that region

during MAM, but after strong AMOC phases. Since this skill increase in spring does not fit the D16 mechanism, it is unlikely to originate from the examined mode of AMOC fluctuations, making room for different mechanisms to be explored and discussed in the future. This finding is supported by the evidence of a strong influence of stochastic atmospheric variability for this region at 2-4 months lag, particularly during spring (cf. Fig. 6 a) and calls for other physical mechanisms, that, if considered in the prediction, could result in a prominent effect on the hindcast skill. Recently, a similar approach invoked a chain of physical

processes in the prediction and achieved improved skill for European summer climate predictions (Neddermann et al., 2018). Additionally, several studies have shown that one of the most robust remote impacts of ENSO is the teleconnection to tropical North Atlantic SSTs in boreal spring (e.g. García-Serrano et al. (2017)). The incorporation of another physical link into the prediction, such as ENSO, could show additional refined information on the North Atlantic SST prediction skill. Our findings therefore illustrate that predicting North Atlantic climate on the seasonal time scale is a complex endeavour with plenty of

possible drivers of skill that a simplified analysis using a single skill precursor such as AMOC cannot fully explain.

Our analyses support further investigation of the AMOC strength and its associated heat transport as complementary information for the seasonal prediction of SSTAs. Both high-resolution coupled models and the two ongoing AMOC monitoring programs RAPID-MOCHA (Cunningham et al., 2007) and OSNAP (Lozier et al., 2017, 2019) are essential for a thorough understanding of the mechanisms analysed here. In fact, the seasonal relationship between AMOC and SSTA could contribute

to the knowledge of the potential applications of a real-time data delivery system, when finally implemented in the RAPID array (Rayner et al., 2016).

## 5 Conclusions

We assess the impact of AMOC fingerprints on North Atlantic seasonal SST variability and predictability across seasons and time. We consider the physical mechanism proposed by D16 in the hindcast skill analysis of a 30-member set of ensemble

hindcasts with the MPI-ESM-MR initialised every February, May, August and November, and evaluate the effect of this mechanism by exploring the changes in SST hindcast skill for tropical and subtropical North Atlantic SST, when compared to the hindcast analysis without considering this mechanism. Our analysis highlights promise, but also numerous subtleties when considering the D16 mechanism for seasonal SST hindcast skill. Specifically, our findings suggest that:

*Variability*

1. For the period of 1979 - 2014, the AMOC strength at 26°N leads a SSTA dipole pattern in the tropical and subtropical North Atlantic with maximum correlations at 2-4 months, in line with the findings of D16 using AMOC observations from RAPID.

2. The seasonal impact of AMOC on SST is accomplished by significant impact of meridional ocean heat convergence. There is also a potential contribution from zonal heat transports.

3. This AMOC fingerprint has a seasonal dependence, and is sensitive to the length of the time window used. This sensitivity affects both the intensity and structure of the fingerprint, which is stronger in spring and summer than during autumn and winter, and over the last decade than for the entire time series back to 1979.

4. The AMOC fingerprint's seasonality can be attributed to i) the influence of stochastic atmospheric variability on SST via atmospheric heat fluxes, which is most pronounced for spring and winter SST variability over the subtropics, weakening the effects of AMOC fingerprint; and ii) Ekman transport, which explains SST variability over the subtropical lobe of the AMOC fingerprint during spring.

*Predictability*

5. Considering D16's physical mechanism in our prediction skill analysis for the period of 1982 - 2014 may result in improved SST hindcast skill for 2-4 months lead time over parts of the subtropical and tropical North Atlantic, mainly during boreal summer (JJA).

    (a) For strong AMOC phases at 26°N

        i. SST hindcast skill for JJA improves over some regions of the subtropics as a result of higher influence of the ocean's thermal memory on SST predictability, following a convergence of OHT north of 26°N.

        ii. During JJA, ACCs increase significantly where they are impacted by AMOC, beating persistence skill as well as skill after weak AMOC and as evaluated for the full time period.

        iii. During winter, spring and autumn, major skill improvement through AMOC is hindered due to limited influence of AMOC (SON), direct influence from the atmosphere (MAM and DJF), prominent influence of Ekman transport on SST (MAM), and potential impact of zonal and vertical oceanic heat transport.

    (b) For weak AMOC phases at 26°N

        i. SST predictive skill is marginally improved over the tropics for JJA SSTAs, as a result of OHT convergence south of 26°N, in agreement with D16's physical mechanism.

        ii. No other seasons or regions show a strong impact of AMOC on seasonal SST hindcast skill in accordance with the D16 mechanism.

We find subtle SST skill increase from AMOC variability according to the D16 mechanism. Only during summer, significant skill increase is found after strong AMOC phases in the subtropics. During other seasons and in other regions, the influence

of the D16 mechanism on seasonal SST prediction skill is limited. While our work brings out both potential and the need for caution when invoking D16's physical mechanism, it highlights a dependence on the AMOC initial state at 26°N when interpreting the mean regional SST skill a season ahead for a particular ensemble prediction system. Given the AMOC initial state, the skill for the mean prediction can therefore represent higher or lower regional SST skill in comparison to the average, which calls for caution when analysing SST predictions.

*Code availability.* Model simulations were performed using the high- performance computer at the German Climate Computing Center (DKRZ). All data are stored at the DKRZ in archive and can be made accessible upon request (www.dkrz.de/up). Analysis was performed using NCAR Command Language packages available at https://github.com/NCAR/ncl.

*Video supplement.* A video supplement related to this paper is available from the repository Technische Informationsbibliothek (TIB) at: https://doi.org/10.5446/50922.

*Author contributions.* JO wrote the article and performed the analysis. MD ran the simulations. LB, AD, JB and JO conceived the study and contributed to the writing and interpretation of the results.

*Competing interests.* The authors declare not to have any competing interests.

*Acknowledgements.* The authors would like to thank Joel Hirschi and Simon Josey for helpful discussions, as well as to the Climate Modelling group at the University Hamburg. This work received funding from the Deutsche Forschungsgemeinschaft (DFG, German Research Foundation) under Germany's Excellence Strategy EXC 2037 'Climate, Climatic Change, and Society' Project Number: 390683824, contribution to the Center for Earth System Research and Sustainability (CEN) of Universität Hamburg (JB). It was further funded by the International Max Planck Research School on Earth System Modelling, by the European Union under Horizon 2020 Project EUCP (Grant Agreement 776613), and by the ANR-TREMPLIN ERC project HARMONY, Grant Agreement number ANR-20-ERC9-0001 (LB). JO was supported by the MER consortium under an Erasmus Mundus scholarship. The model simulations were performed using the high-performance computer at the German Climate Computing Center (DKRZ).

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
