# Peer review of "Subtle influence of AMOC on seasonal SST hindcast skill in the North Atlantic"

_Weather and Climate Dynamics, 2020_

## Referee Comment (RC1) · Anonymous Referee #1 · 20 Oct 2020

Review of manuscript wcd-2020-45 submitted to Weather and Climate Dynamics

AMOC fingerprints influence seasonal SST predictability in the North Atlantic

by Oliveira et al.

This study investigates the impact of the strength of the AMOC on the seasonal prediction skill of North Atlantic SSTs using the MPI-ESM. The authors find that, consistent with previous work (Duchez et al. 2016), the AMOC leads a dipole SST pattern in the tropical and subtropical North Atlantic. The study then assesses whether considering different phases of the AMOC leads to improved seasonal prediction skill of SST.

In general, I find the study interesting, and the results should be of interest to the readers of Weather and Climate Dynamics and especially to those in the emerging

field of climate prediction. However, I believe this submission requires major revisions before beings suitable for publication, as outlined below.

Primary concerns:

1. The study relies on the mechanism of Duchez et al. (2016) (D16), and much of the analysis is a repeat of that done in D16 using the MPI-ESM rather than observations. The mechanism suggests that (l.36-37) "a stronger than average AMOC at 26N advects more heat northward, leading to colder waters in the tropics and warmer water in the subtropics." This might be (and probably is) true, however no analysis is presented relating the SST variability and predictability to changes in ocean heat transport (OHT) convergence, and the analysis is only based on the correlation between SST and AMOC. In the model, the importance of OHT to SST variability could easily be quantified (for example by upper-ocean budgets for Box 1,2). This would also quantify the role of surface heat fluxes (as discussed in the manuscript). Similarly, a more quantitative analysis could be performed to firmly establish the link between the proposed mechanism and SST prediction skill (e.g., Yeager et al. 2020, https://doi.org/10.1007/s00382-020-05382-4). In summary, since much of this study repeats the analysis of D16, I think the authors should use the opportunity and tool available to add more to the understanding of the identified SST prediction skill.

2. Related to the above, I believe the discussion of the new findings could be improved and more balanced. In places, I find the discussion a bit "selective", i.e., focusing on where the results fit with the initial hypothesis (e.g., l.245-246). It would perhaps help if the difference in skill for different regions (Box1,2?) was highlighted in a separate figure. Also, what are the confidence intervals of the correlations in Fig.8; are the difference in skill significant?

3. The text is in many places quite hard to follow (see specific comments below), and I think the authors should spend some time/effort in improving the general flow of the text.

[Figure]

Specific comments

l.3 I don't think it should be necessary for the reader to be familiar with D16 to read the abstract. I think you should rather briefly explain the mechanism.

l.18 change to "have potential important socio-economic"?

l.21-40 I think the authors should revisit the structure of these paragraphs. A suggestion would be to do l.24-33, then 21-23, and then 34-40.

l.21 D16's mechanism -> The mechanism in D16

l.21 transition -> variations?

l.24 "fluctuations in the atmosphere" – could you be more specific? E.g., atmospheric circulation? Also, are you trying to say that ASFs and Ekman-induced heat transport by atmospheric circulation variability are important to SSTs, or that "fluctuations in the atmosphere" are an additional driver?

l.27 "been additionally" -> also

l.42 "two dominant mechanisms" – could you be more specific?

l.47-49 It is not clear how this paragraph relates to the previous.

l.51 "similar technique as Borchert et al. (2018)" – please elaborate

l.64 "in its mixed resolution" – I am not sure what this means. Also, check sentence.

l.81 Statistical Methods should include a description of the significance test used (bootstrapping).

l.83-85 check sentence

l.90 (and in general) the manuscript contains numerous abbreviations. I think the text would be improved if the use of abbreviations was somewhat limited.

l.93: "fluxes over sea" -> fluxes over the ocean

l.99-100 is it necessary to use a 3-month running average when you work with seasonal means? Also, "high frequency" is a relative term. What is the "high frequency" variability that you want to remove?

l.109 verify -> evaluate

l.109 change to "against observations... in our analysis"?

l.110 "Statistics" - could you be more specific?

l.113 when is the smoothing applied? The gray lines in Fig.1 look unfiltered. And again, why is it necessary to display and evaluate a smoothed seasonal cycle? Minimum and maximum values are also given for individual months.

l.113 check use of "c.f" throughout the text. I don't think this is the appropriate use.

l.119 remove "lower"

l.122 "and correlate with" -> "with a correlation of"

l.125 check sentence

l.135 any western displacement is not easy to see. Also, to me it looks like the eastern displacement of maximum correlation between lag 4 to 7 stretches northeastward toward the coast of Spain/northern Africa. Is this consistent with advection by the subtropical gyre?

l.152 you could consider adding a panel to Fig.4 showing this for e.g., Box 1,2.

l.155 It is true that Fig.4 shows time series and correlations from box 1,2, but is it correct to say that "main spatial features" are displayed in Fig.4?

l.174 I'm not sure I see why positive correlations over the subpolar region necessarily point to atmospheric forcing (without any additional analysis). Please elaborate.

l.182 Again, is atmospheric forcing the only other option? What about other oceanic forcing not captured by the AMOC?

l.185-198 The analysis/discussion of ASFs and SST is based entirely on correlations, but nothing is said about the magnitude of the anomalous ASFs, and, hence, how much of the SST variability they are responsible for.

l.196 "(Fig. 6e,f) - strong positive correlations only seen in (f)

l.196 it is not easy to see from Fig. 6f how much the positive correlations overlap with the subtropical lobe of the AMOC fingerprint. To me it looks like the positive correlations are mainly further east. Please make clearer and/or quantify.

l.201 "internal AMOC signal" - please explain

l.206-207 There are almost no significant correlations in the tropical lobe for AMOC-EKM at 2-4 month lag (Fig.7). So not sure I understand this sentence.

l.204-205 "the AMOC-SST dipole for both autumn and winter" - on l.172 you state that there is no distinct dipole pattern in autumn and winter.

l.232 do your results change if you only consider AMOC stronger/weaker than e.g., 1 std

l.233 "Atlantic Meridional Variability" - do you mean Atlantic Multidecadal Variability"? In any case, please explain/define AMV.

l.237-241 This is where I think the authors need to strengthen their analysis, to demonstrate that the mechanism outlined here is actually what carries the added prediction skill.

l.249 where is the MDR?

l.266 no analysis of heat advection is performed

l.272 Although decadal AMOC variability influences the AMV, I don't think it's correct to say "i.e., AMV".

l.273 Related to l.272, is the modeled AMOC also anomalously strong for the RAPID

period (corresponding to a positive AMV phase)?

l.274 "multidecadal" - on l.272 you were talking about "decadal"

l.297 "only explaining the improvements over the subtropics" - it is not easy to see the skill improvements in MAM, especially for the subtropics. As mentioned above It would perhaps help if the difference in skill for different regions (Box1,2?) was highlighted in a separate figure, including confidence intervals of the correlations.

l.298 check sentence

l.309-311 And what would a model heat budget say?

l.311 "active ocean dynamics" - please elaborate (what would inactive dynamics be?). Also does "active ocean dynamics" include Ekman-driven heat transport?

l.352 "following a convergence of OHT" - not shown

l.355 "large parts of the subtropics" - not sure I agree if this mainly refers to the afore-mentioned area around 40W.

Figure 4. Add significance to (b)

Figure 6. Figure title says "cumulated". Should be "cumulative"?
* * *

---

## Referee Comment (RC2) · Anonymous Referee #2 · 2 Nov 2020

Review of "AMOC fingerprints influence seasonal SST predictability in the North Atlantic", by Oliveira et al.

General Comments:

This manuscript presents an analysis of the impact of the AMOC strength at 26N on the North Atlantic SST (based on Duchez et al., 2016) in the MPI-ESM-MR seasonal forecast system. First the authors evaluate the AMOC strength and fingerprints in an assimilation experiment ran with MPI-ESM-MR (used as initial conditions for the seasonal forecasts) and show similarities with the results of Duchez et al. (2016). The authors then compute the AMOC fingerprints for each season, which is the novelty of the paper, and investigate the impact of the atmospheric forcing (air-sea fluxes and Ekman-induced heat transport) on the SST variability which may be dominant on sea-

sonal time-scales. The authors then evaluate and compare the skill of North Atlantic SST in the MPI-ESM-MR seasonal forecast system to determine whether considering strong and weak AMOC phases results in improved skill.

The quality of the writing is good overall but some sections are hard to follow the clarity should be improved (see detailed comments below). The analysis and results are interesting, and once the manuscript is improved (especially the discussion of the results) I believe it would be of interest to the seasonal prediction community. I therefore suggest this paper to be considered for publication after major revision.

Specific Comments:

1) I believe it would help the clarity of the manuscript if in each section of the paper it would be clearly stated which experiment is being analysed, as it took me some time to figure out if it was the assimilation experiment or the seasonal predictions. This should also be indicated in the figures, for example in the caption of figure 1, 'The AMOC in MPI-ESM-MR' is confusing as it does not refer to the experiment type. It would also help to motivate in the text why the experiment being used is suitable for the analysis. Since the first part of the paper considers only the assimilation experiment and it is constraint to observations, do you expect the conclusion to hold for the seasonal forecasts? The authors could consider computing the AMOC fingerprints with the seasonal forecasts.

2) The main concern I have with the paper is the interpretation of the seasonal SST skill results in terms of D16's mechanism for the strong and weak AMOC phases (section 3.3). I find the discussion insufficient to conclude whether the SST skill results are consistent D16's mechanism.

Comments by line:

Line 16: 'tropical seasonal SST anomalies' ->'Seasonal SST anomalies in the tropics'?

Line 21: Could explain briefly D16's mechanism to make the paper self sustained,

having only to refer to D16 for specific details.

Line 21: 'links transition' -> 'links the transition'?

Lines 29-32: These sentences could be made clearer to make it easier to follow.

Lines 41-46: This paragraph could also be made clearer. The authors could explain what is meant by 'incorporating known physical mechanisms into their seasonal prediction analysis'. From the latter sentences I infer it means through the initial conditions? Perhaps the physical mechanisms could be briefly mentioned.

Line 47: The term 'potential predictability' could be briefly explained. Does it refer to the ability of the model to predict itself?

Line 49: Does 'initialised coupled simulations' refer to a set of hindcast (retrospective forecasts)? If so I would use this term to be more precise. The authors could briefly explain the evaluation technique in Borchert et al. (2018) to make the paper self sustained.

Line 71: This sentence should written more precisely. 'full-field' initialisation refers to the initialisation method of the forecast system. While nudging refers to assimilation method used in the reconstruction where the initial conditions are taken from. The assimilation experiment is fully coupled right?

Line 98: Why is a linear trend removed?

Line 108: 'Model verification for AMOC' -> 'Verification of the AMOC in the assimilation experiment' or 'Verification of the AMOC in the reconstruction'?

Line 109: 'to' -> 'with', 'to' -> 'in'.

Lines 109-110: I don't think these sentences are necessary. As already mentioned I find the sentences unclear as the experiment analysed it not mentioned.

Table 1 is not discussed, so either comment on it or remove the table. What does

'seasonal range' refer to? Should be defined.

Line 112: To compute the anomalies, why is the annual mean of each year removed rather than the annual climatology?

Lines 111-122: In this paragraph figures 1b,d and f are not referred in the text or discussed, so are they necessary? Also the figures are referred in the text with 'c.f.' when it is not necessary. The clarity of the paragraph could be improved. In lines 119-120 the 'seasonal range' is commented but no reference to table 1 is provided.

Figure 1: The labelling of the panels in the bottom right corner instead of in the top left may be slightly confusing, although a minor detail. Is AMOC-EKM (red line) necessary in the figure 1g?

Line 127: 'model' -> 'assimilation experiment'

Lines 125-142: The paragraph structure and the clarity of this section should be improved.

Lines 126-130 the authors describe the lagged correlation patterns between the AMOC and the SSTA as being the same for all lags. However, to me it seems that the pattern correlation evolves from lag 0 to month 7, as later described by the authors, which seems contradictory. I believe this should to clarified.

Line 145: Could reference Fig. 2 in this sentence.

Line 152: 'not shown', could show in the supplementary material.

Line 155: The first sentence seems unnecessary.

Lines 159-164: This paragraph seems a repetition of the previous paragraphs.

Lines 155-159: Related to a previous comment, a limitation with the definition of box 2 (Fig. 4) is that the positive correlation at later lags shifts towards the Eastern North Atlantic which is not included in the box. This indicates that the lag correlation patterns

are not constant.

Line 166: delete 'here'.

Lines 166-175: It is not clear what period is being used, I suppose that the 30-year period?

Line 174: Why atmospheric drivers? Is this the only option? In fact, this seems inconsistent with the results shown in the next section (Fig. 7m).

Line 177: 'is response' -> 'is the response' or 'responds'

Lines 176-213: The clarity of this section could be improved and perhaps made shorter. I believe this section could also be improved by commenting on the implications of the results.

Line 218: 'The hindcasts' -> 'The hindcasts initialised in'.

Line 222: I assume that AUG and MAY refer to JJA and SON in fig. 8, if so this makes the text confusing, it would help to refer to the season like in fig. 8.

Line 229: 'The role of AMOC' -> 'The role of the AMOC'.

Line 231: What does 'separately' mean here? Is there any difference with the previous section?

Line 233: Does this mean that the startdates are divided into two sets, strong and weak AMOC?

Line 233 'Atlantic Meridional Variability (AMV)' -> 'Atlantic Multidecadal Variability (AMV)'?

Line 238: 'sub-' -> 'sub-tropical'

Line 242: 'we find higher hindcast skill for DJF...' this statement is not so clear to me. The magnitude of the skill in some regions seems higher, but the area of skill seems smaller.

Lines 237-258: These paragraphs constitute some of the main results of the paper, but they seem insufficient to me and hard to follow. The interpretation of D16's physical mechanism in terms of skill does not seem very clear and I struggle to follow the interpretation of the results. Also, based on the AMOC seasonal fingerprints I would expect that D16's mechanism could perhaps explain the SST skill in spring and maybe in summer, but not autumn or winter.

Line 241: The statement: 'For strong AMOC phases, we find higher hindcast skill for DJF, JJA and MAM SSTAs over the subtropics in comparison to ACCs considering the full period', I am not certain about this statement. Comparing the skill maps it seems that the magnitude of the skill is greater in some regions but the areas with positive skill smaller. Perhaps computing maps of the difference of correlation (including the significance) could help the interpretation. Could the SST skill maps be affected by the smaller sample size when comparing the entire period and positive and negative AMOC composites?

Line 296: 'a more active ocean'? what does this refer to?

---

## Author Comment (AC1) · 1 Feb 2021

**Response to Referee#1**

We thank the anonymous referee for their valuable comments and constructive reviews. These comments certainly help us to identify the sections in the manuscript that required improvement. We highlight the major improvements as follows:

- We carried out an analysis of ocean heat transport convergence in two regions covering the tropical and the subtropical North Atlantic, adding two new panels for Fig. 4;
- We included a more detailed comparison of statistical significance of skill differences given the AMOC state, adding a new figure (Fig. 9);
- We rewrote major portions of the manuscript to make it clearer and improve the reading flow.

We took into consideration all suggestions made by the reviewer and we provide below a point-by-point response to each comment. Please note that the referee's comments are highlighted in **bold** font, while our answers are in regular font.

**Primary concerns:**

**1. The study relies on the mechanism of Duchez et al. (2016) (D16), and much of the analysis is a repeat of that done in D16 using the MPI-ESM rather than observations. The mechanism suggests that (l.36-37) "a stronger than average AMOC at 26N advects more heat northward, leading to colder waters in the tropics and warmer water in the subtropics." This might be (and probably is) true, however no analysis is presented relating the SST variability and predictability to changes in ocean heat transport (OHT) convergence, and the analysis is only based on the correlation between SST and AMOC. In the model, the importance of OHT to SST variability could easily be quantified (for example by upper-ocean budgets for Box 1,2). This would also quantify the role of surface heat fluxes (as discussed in the manuscript). Similarly, a more quantitative analysis could be performed to firmly establish the link between the proposed mechanism and SST prediction skill (e.g., Yeager et al. 2020, https://doi.org/10.1007/s00382-020-05382-4). In summary, since much of this study repeats the analysis of D16, I think the authors should use the opportunity and tool available to add more to the understanding of the identified SST prediction skill.**

We thank the reviewer for the suggestions. The panels added to Fig. 4 illustrate that oceanic heat convergence in longitudinal bands north and south of the AMOC latitude (defined as total heat inflow at the southern boundary minus total heat outflow at the northern boundary) match qualitatively the seesaw mechanism from D16, with increased heat convergence south of 26N after weak AMOC (indicated by a negative correlation to AMOC),

and increased heat convergence north of 26N after strong AMOC (i.e. positive correlation to AMOC). The imperfect correlation values indicate limitations of our analysis, which we now explicitly discuss in several sections of the manuscript (most prominently the discussion section). In essence, this analysis highlights the complexity of factors influencing North Atlantic SST, which we cannot fully capture with the simplified analysis performed here. While this clearly is a drawback of our analysis, the newly added Fig. 9 illustrates that significantly distinct skill estimates are obtained after strong and weak AMOC phases in both boxes during summer, so our mechanism appears to capture the dynamics sufficiently well to yield discernible skill estimates, illustrating its value.

**2. Related to the above, I believe the discussion of the new findings could be improved and more balanced. In places, I find the discussion a bit "selective", i.e., focusing on where the results fit with the initial hypothesis (e.g., l.245-246). It would perhaps help if the difference in skill for different regions (Box1,2?) was highlighted in a separate figure. Also, what are the confidence intervals of the correlations in Fig.8; are the difference in skill significant?**

In response to this comment, we carefully revised the discussion and made an effort to balance the text. In particular: i) we included remarks on the role of ocean heat transport convergence to the referred regions (Box1 and 2), ii) we included a new figure (Fig. 9) to illustrate the difference in skill (Box 1 and 2). This analysis illustrates that skill differences are significant during certain seasons, which is in line with the D16 mechanism (as discussed in the manuscript).

**3. The text is in many places quite hard to follow (see specific comments below), and I think the authors should spend some time/effort in improving the general flow of the text.**

We put a major effort in improving the text of the entire manuscript, which will hopefully become clear both in the responses to the reviewer's specific comments and the revised manuscript.

**Specific comments**

**l.3 I don't think it should be necessary for the reader to be familiar with D16 to read the abstract. I think you should rather briefly explain the mechanism.**

To make the manuscript self-sustained, we changed the sentence as follows: *'We test the dependence of SST predictive skill in initialised hindcasts on the phase of AMOC at 26°N, invoking a seesaw - like mechanism driven by AMOC fluctuations.'*

**l.18 change to "have potential important socio-economic"?**

We changed the text to: *'Seasonal SST anomalies (SSTAs) in the tropics have been linked to the intensity and genesis of tropical cyclones and heatwaves (Coumou and Rahmstorf, 2012; Duchez et al., 2016b; Arora and Dash, 2016), and to fluctuations of marine resources (Stock et al., 2015); all of which have potential important socio-economic consequences.'*

**l.21-40 I think the authors should revisit the structure of these paragraphs. A suggestion would be to do l.24-33, then 21-23, and then 34-40.**

We agree with the referee's suggestion and changed the structure as suggested.

**l.21 D16's mechanism -> The mechanism in D16**

We agree and we modified these lines as follows: *'Here, we examine the seesaw mechanism proposed by Duchez et al. (2016a) (henceforth D16), which links variations in strength of the Atlantic Meridional Overturning Circulation (AMOC) at 26N and North Atlantic SSTs on monthly time scales.'*

**l.21 transition -> variations?**

And stated in the previous response this line was modified as follows: *Here, we examine the seesaw mechanism proposed by Duchez et al. (2016a) (henceforth D16), which links variations in strength of the Atlantic Meridional Overturning Circulation (AMOC) at 26N and North Atlantic SSTs on monthly time scales.'*

**l.24 "fluctuations in the atmosphere" – could you be more specific? E.g., atmospheric circulation? Also, are you trying to say that ASFs and Ekman-induced heat transport by atmospheric circulation variability are important to SSTs, or that "fluctuations in the atmosphere" are an additional driver?**

Thanks for the question. We specifically refer to ASFs and Ekman-induced heat transport in the ocean, and their role in modulating SSTs. Therefore we rewrote the sentence as follows: *'Air-sea heat fluxes (ASFs) and Ekman-induced oceanic heat transport are important drivers of seasonal variability for SSTs (Bjerknes 1964).'*

**l.27 "been additionally" -> also**

We changed as suggested: *'Part of the North Atlantic seasonal SST variability has also been attributed to the AMOC (e.g. Bryden et al. (2014); Zhang et al. (2019)).'*

**l.42 "two dominant mechanisms" – could you be more specific?**

This entire paragraph was rewritten to improve clarity. We change the sentences as follows: *'Recent studies have found improved hindcast skill in the North Atlantic region after considering known physical mechanisms into their seasonal prediction analysis. Mechanisms were invoked in two possible ways: by identifying and explaining times of low and high skill, including precursors of high skill, as so-called* windows of opportunity *(Borchert et al., 2018, Mariotti et al. 2020); or by establishing physical mechanisms in the hindcast ensemble by sub-selecting ensemble members that meet certain physical criteria, thus filtering atmospheric noise in the ensemble (Dobrynin et al., 2018, Neddermann et al., 2018). The present study focuses on oceanic processes that are arguably less noisy than atmospheric dynamics (Gulev et al. 2013).*

**l.47-49 It is not clear how this paragraph relates to the previous.**

In order to clarify, several modifications were done to the introduction text, with the referred lines now in lines 54-57. We hope that the transitions are now clearer.

**l.51 "similar technique as Borchert et al. (2018)" – please elaborate**

Adding detail here, we modified the sentences as follows: *'Analysing an ensemble of yearly initialised hindcasts with MPI-ESM-LR covering 1901-2010, Borchert et al, 2018, Borchert et al, 2019 showed that the AMOC at 50°N influences the SST variability and predictability for several years, with higher skill after years of strong AMOC and vice versa. Borchert et al, 2018 perform a predictive skill analysis of SST conditioned to strong and weak OHT anomalies at 50°N separately, showing a robust influence of the ocean on windows of opportunity for decadal subpolar North Atlantic SST predictions.'*

**l.64 "in its mixed resolution" – I am not sure what this means. Also, check sentence.**

In MPI-ESM-MR, MR stands for mixed resolution: T63 with 95 levels in the atmosphere and 0.4° horizontal resolution with 40 levels in the ocean. This information is included in the manuscript in lines 74-79.

**l.81 Statistical Methods should include a description of the significance test used (bootstrapping).**

The manuscript now includes a description of the bootstrapping method in lines 128-130 as follows: *'We calculate statistical significance using a Monte-Carlo bootstrapping method. The process consists of 1000 bootstraps with replacement on the time-dimension at the 95% confidence level.'*

**l.83-85 check sentence**

We rewrote the sentence as follows: *'We choose the assimilation experiment over observations because of the short observational record of AMOC from the RAPID/MOCHA array that is available only from April 2004 (Cunningham et al. 2007). Our method therefore allows to constrain the seasonal cycle more robustly.'*

**l.90 (and in general) the manuscript contains numerous abbreviations. I think the text would be improved if the use of abbreviations was somewhat limited.**

Thanks for the suggestion. We changed the sentence as follows: *'We evaluate the atmospheric contribution to the SST variability using the Ekman transport (EKM) and air-sea heat fluxes.'* We additionally remove from the manuscript the following abbreviation: MDR (hurricane main development region), and made sure to avoid agglomerations of abbreviations in the entire manuscript.

**l.93: "fluxes over sea" -> fluxes over the ocean**

Changed as requested.

**l.99-100 is it necessary to use a 3-month running average when you work with sea sonal means? Also, "high frequency" is a relative term. What is the "high frequency" variability that you want to remove?**

Thanks for your comment. We understand that this sentence was misleading, as we meant applying the low-pass filter only for plotting time series (e.g. Fig. 1, 4) , but not for any analysis of seasonal means. We opted for removing this sentence, since a description of the running average used can be seen in the respective figure caption.

**l.109 verify -> evaluate**

Thanks for your input regarding lines 109 and 110. We decided to delete these lines to improve the reading flow following a suggestion from referee#2.

**l.109 change to "against observations... in our analysis"?**

As stated in the comment above, we deleted these lines to improve the reading flow following a suggestion from referee#2.

**l.110 "Statistics" - could you be more specific?**

As stated in the comment above, we deleted these lines to improve the reading flow following a suggestion from referee#2.

**l.113 when is the smoothing applied? The gray lines in Fig.1 look unfiltered. And again, why is it necessary to display and evaluate a smoothed seasonal cycle? Minimum and maximum values are also given for individual months.**

The smoothing is applied to the grey lines in Fig.1, and not for the mean seasonal cycle (coloured lines). The grey lines in Fig. 1.a-f represent the variability within each analysed year. To make this sentence clearer we rephrased as follows: *'To show the spread of the annual climatology, grey lines in Fig.1.a,c,e represent anomalies w.r.t. the mean transport of a given year calculated for the full time series (1979-2014), and smoothed with a 3-month running average.'*

**l.113 check use of "c.f" throughout the text. I don't think this is the appropriate use.**

We agree and removed the abbreviation in the manuscript where inappropriate.

**l.119 remove "lower"**

Changed as suggested.

**l.122 "and correlate with" -> "with a correlation of"**

Changed as suggested.

**l.125 check sentence**

We rephrased the sentence as follows: *'Here, we compare the observed AMOC fingerprints discussed in D16 with those present in the assimilation experiment for the period April 2004 to March 2014 (c.f. D16's Fig.3).'*

**l.135 any western displacement is not easy to see. Also, to me it looks like the eastern displacement of maximum correlation between lag 4 to 7 stretches northeastward toward the coast of Spain/northern Africa. Is this consistent with advection by the sub-tropical gyre?**

We agree that this needed rephrasing to point out more specifically how the northern lobe behaves, and we altered the sentences as follows: *'With increasing time lag (5-7 months specifically), the subtropical lobe of positive correlation shows a displacement towards the east along the approximate circulation path of the northern boundary of the subtropical gyre. This suggests a role for the subtropical gyre in advecting the northern lobe of the seasonal AMOC fingerprint eastward.'*

**l.152 you could consider adding a panel to Fig.4 showing this for e.g., Box 1,2.'**

Thanks for your suggestion. We included one animation as supplementary information which illustrates the time dependence discussed in this paragraph.

**l.155 It is true that Fig.4 shows time series and correlations from box 1,2, but is it correct to say that "main spatial features" are displayed in Fig.4?**

We agree that this needed rephrasing and removed the sentence.

**l.174 I'm not sure I see why positive correlations over the subpolar region necessarily point to atmospheric forcing (without any additional analysis). Please elaborate.**

We agree that this sentence needed rephrasing to bring out our point more clearly. We elaborate on the possibility of atmospheric contribution in section 3.2.4. We changed to: 'In contrast, we find that autumn and winter seasons lack a characteristic dipole pattern (Fig. 5.c,d), showing instead only a narrow region of negative correlations over the subtropics of the order of -0.2 (-0.1) for winter (autumn). The absence of a dipole pattern in autumn and winter may suggest the influence of atmospheric drivers that could potentially supersede the AMOC fingerprints during these seasons.'

**l.182 Again, is atmospheric forcing the only other option? What about other oceanic forcing not captured by the AMOC?**

As pointed out by the reviewer, it is entirely possible that other oceanic forcings contribute to seasonal SST variability as well, such as zonal or vertical heat advection in the ocean, or heat storage effects. Such contributions are not captured by AMOC variability. Since the aim of this work is to discern the AMOC effect on predictability, and the main counteracting factors come from the atmosphere (other oceanic drivers would likely contribute positively to predictability), we stand by the analysis we did. That being said, we changed the phrasing slightly so as to leave room for the other possible forcings to SST, as pointed out by the reviewer.

**l.185-198 The analysis/discussion of ASFs and SST is based entirely on correlations, but nothing is said about the magnitude of the anomalous ASFs, and, hence, how much of the SST variability they are responsible for.**

This is true and a valid concern raised by the reviewer. In this work, we consider that ASFs are strong drivers of SST whenever they significantly contribute to SST variability. This is measured by significant correlation between the two quantities. It is not our aim to perform a full heat budget which would require a quantification of the total contribution of ASFs to SST variability, mainly because such budgets are rarely closed in climate models to begin with. To make the aims of our analysis clearer, we rephrased this section and most prominently replaced "strong" with "significant" when writing about the ASF influence on SST variability.

**l.196 "(Fig. 6e,f) - strong positive correlations only seen in (f)**

We agree with the reviewer and corrected the reference to Fig. 6: *'In contrast, we note strong positive ASF-SST correlations 12 over the eastern part of the subtropical lobe of the AMOC-SST dipole for autumn (Fig.6f).'*

**l.196 it is not easy to see from Fig. 6f how much the positive correlations overlap with the subtropical lobe of the AMOC fingerprint. To me it looks like the positive correlations are mainly further east. Please make clearer and/or quantify.**

The text now specifies that this finding is mostly limited to the eastern part of the subtropical lobe of the AMOC fingerprint.

**l.201 "internal AMOC signal" - please explain**

We used the quantity 'AMOC-EKM', i.e. subtracting the Ekman transport from AMOC, to remove the short-term variability attributed to the Ekman component (e.g. Mielke et al. 2013). This leaves the non-Ekman part of the circulation, which we call the 'internal signal' of AMOC. Due to the confusion that this phrasing has caused, we removed it from the manuscript. For clarity, we rephrase this as follows: *'In addition to ASFs, Ekman transport is an important contributor to short-term SST variability (Frankignoul, 1985). EKM is the wind-driven component of the overturning in the ocean, forming the full AMOC signal together with the overturning in the ocean interior, to which usually most of the northward heat transport is attributed (Ferrari and Ferreira, 2011).'*

**l.206-207 There are almost no significant correlations in the tropical lobe for AMOC-EKM at 2-4 month lag (Fig.7). So not sure I understand this sentence.**

This sentence was indeed misleading, referencing a correlation analysis further out than 4 months that we conducted but do not show. To make the paper more succinct and to the point, we deleted this sentence.

**l.204-205 "the AMOC-SST dipole for both autumn and winter" - on l.172 you state that there is no distinct dipole pattern in autumn and winter.**

We agree that this needed rephrasing. We modified as follows: *'In the other seasons, SST variability seems to be less less influenced by EKM, as shown by the weak correlation pattern for EKM (Figs.7b,l) as well as the high similarity in the SST patterns for AMOC and AMOC-EKM (Figs.7a, c, j, m).'*

**l.232 do your results change if you only consider AMOC stronger/weaker than e.g., 1Std**

The essence of our results do not change when using 0.5 std as selection criterion (please see example below, Fig.2), and also remain robust when using 1 std. However, to avoid shortening the time series too much, we choose to consider positive and negative anomalies instead. We now mention this in the manuscript to illustrate the robustness of our findings (beginning of the section "The role of the AMOC").

[Figure]

Fig. 2: Similar to Fig. 8 in the manuscript, but considering strong/weak phases of AMOC for a 0.5 std threshold.

**l.233 "Atlantic Meridional Variability" - do you mean Atlantic Multidecadal Variability"?**
**In any case, please explain/define AMV.**

While rewriting, we decided to remove this sentence from the manuscript to make the discussion more focused on the AMOC strength as the chosen criteria. For the record, this is indeed what we meant.

**l.237-241 This is where I think the authors need to strengthen their analysis, to demonstrate that the mechanism outlined here is actually what carries the added prediction Skill.**

In rewriting and streamlining this manuscript, we moved this paragraph to the variability section. Please see our response to 'Primary concerns 1.' in the beginning of this document.

**l.249 where is the MDR?**

The hurricane main development region (MDR) is located in the tropical North Atlantic: 10–20°N, 30–60°W. We added this information and modified the sentence: '*While the mechanism does not solely explain the hindcast skill behaviour in the tropics, we find an improvement over the hurricane main development region, 10–20°N, 30–60°W, (e.g. Hallam et al. 2019) (...).*

**l.266 no analysis of heat advection is performed**

Please note that we now include an analysis of heat transport convergence associated with the AMOC in section 3.2.2 (Fig. 4.e,f) in lines 196-210.

**l.272 Although decadal AMOC variability influences the AMV, I don't think it's correct to say "i.e., AMV".**

We agree, thanks for the remark. We rewrote the sentence as follows: 'A possible reason for these differences could be decadal changes in AMOC variability and their imprint on SST (e.g. Ba et al. (2014); Knight et al. (2005)).'

**l.273 Related to l.272, is the modeled AMOC also anomalously strong for the RAPID period (corresponding to a positive AMV phase)?**

Observed and modelled AMOC are in reasonable agreement during the RAPID period (see fig. 1g), both in terms of variability and mean state. This statement is therefore true for both model and observations, and we added a short comment on that in place.

**l.274 "multidecadal" - on l.272 you were talking about "decadal"**

The AMOC changes referenced here are closer to a multidecadal mode than a decadal one. We adopted the use of "multidecadal" here and before.

**l.297 "only explaining the improvements over the subtropics" - it is not easy to see the skill improvements in MAM, especially for the subtropics. As mentioned above It would perhaps help if the difference in skill for different regions (Box1,2?) was highlighted in a separate figure, including confidence intervals of the correlations.**

Thanks for your suggestion. We include a new figure in the manuscript (Fig. 9) to illustrate the difference in skill for Box 1 and 2. This figure alongside new discussion of its contents now illustrates that the main season for skill improvement through the D16 mechanism is summer. This is reflected in the manuscript, also conveying more nuance when discussing the MAM fingerprint.

[Figure]

*'Fig. 9. SST ACCs against ERA-Interim at 2-4 months lead time averaged over the regions shown in Fig. 4a. a) Box 1 (10.5° - 22.5°N, 22° - 55°W), and b) Box 2 (28.5° - 40.5°N, 40° - 70°W. Black lines represent the ACCs considering the full time series (1982-2014), red lines for strong, and blue lines for weak AMOC phases. The shaded areas indicate the interquartile ranges.'*

**l.298 check sentence**

We entirely rewrote the paragraph. This led to a dissolution of this sentence into several new sentences, which hopefully clarified this issue.

**l.309-311 And what would a model heat budget say?**

In order to streamline the paper, we removed the entire paragraph as it did little to discuss the presented findings. As to the reviewer's question, we can only speculate what a careful model-based ocean heat budget would say, but from the heat convergence analysis added

to the paper (Fig. 4) and based on some work done by Kröger et al. (2018) on the subpolar gyre in a similar model setup, we suspect that a model heat budget would agree with Roberts et al. (2017) in that the ocean drives midlatitude heat content changes.

**l.311 "active ocean dynamics" - please elaborate (what would inactive dynamics be?). Also does "active ocean dynamics" include Ekman-driven heat transport?**

'Active' here refers to the oceanic response in contributing to the variability of temperatures in the surface mixed layer, as opposed to a 'passive' response to local atmospheric forcings. In the context of Roberts et al. (2017), 'active ocean dynamics' includes Ekman-driven heat transport. Please note, however, that when revising the manuscript we came to delete this sentence.

**l.352 "following a convergence of OHT" - not shown**

In response to the reviewer's comments, we added an analysis of oceanic heat convergence to the manuscript (Fig. 4). This part is therefore now explicitly assessed.

**l.355 "large parts of the subtropics" - not sure I agree if this mainly refers to the aforementioned area around 40W.**

We deleted "large", as we agree that this was an overstatement of our findings.

**Figure 4. Add significance to (b)**

We added the significance as suggested.

**Figure 6. Figure title says "cumulated". Should be "cumulative"?**

Based on published literature, we believe that both forms can be used (e.g. Duchez et al. Fig. 8) and we prefer to keep "cumulated" as it is.

**References**

Mielke, C., Frajka-Williams, E., & Baehr, J. (2013). Observed and simulated variability of the AMOC at 26 N and 41 N. Geophysical research letters, 40(6), 1159-1164.

Kröger, J., Pohlmann, H., Sienz, F., Marotzke, J., Baehr, J., Köhl, A., ... & Müller, W. A. (2018). Full-field initialized decadal predictions with the MPI earth system model: An initial shock in the North Atlantic. Climate Dynamics, 51(7-8), 2593-2608.

---

## Author Comment (AC2) · 1 Feb 2021

**Response to Referee #2**

We thank the anonymous referee for their valuable comments and constructive reviews. These comments certainly help us to identify the sections in the manuscript that required improvement. We highlight the key points of this revision as follows:

- We improved the discussion of the role of D16's mechanism on SST predictive skill and included a more detailed comparison of statistical significance of skill differences, adding a new figure (Fig. 9);
- We carried out an additional analysis of ocean heat transport convergence in two regions covering tropical and subtropical North Atlantic, adding two new panels in Fig. 4;
- We show a comparison of the AMOC fingerprints between assimilation experiment and hindcasts;
- We carefully indicated in each section and caption which experiment is being addressed;
- We performed a major revision of the text in the entire manuscript to make it clearer and improve reading flow.

We took into consideration all suggestions made by the reviewer and a point-by-point response to each comment is reported below. Please note that the referee's comments are highlighted in **bold** font, while our answers are in regular font.

**Specific Comments:**
**1) I believe it would help the clarity of the manuscript if in each section of the paper it would be clearly stated which experiment is being analysed, as it took me some time to figure out if it was the assimilation experiment or the seasonal predictions. This should also be indicated in the figures, for example in the caption of figure 1, 'The AMOC in MPI-ESM-MR' is confusing as it does not refer to the experiment type. It would also help to motivate in the text why the experiment being used is suitable for the analysis. Since the first part of the paper considers only the assimilation experiment and it is constraint to observations, do you expect the conclusion to hold for the seasonal forecasts? The authors could consider computing the AMOC fingerprints with the seasonal forecasts.**

We thank the reviewer for raising these points. We agree that the indication of the experiment was not obvious at times, and we now carefully revised the manuscript and made sure to explicitly state which experiment was used in each session and figure caption. Regarding the reviewer's second comment, in Fig. 1 of this document we compare the AMOC fingerprints for the assimilation experiment (Fig. 5 in the manuscript) and the seasonal forecasts. Despite differences in the magnitude of correlations, we find reasonable similarity between AMOC fingerprints for the ensemble mean and assimilation, with overall

agreement in the sign of the correlations. We are therefore confident that our conclusions for sections 3.1 and 3.2 using the assimilation experiment can be extended to section 3.3, where we analyse seasonal forecasts. We add a comment on that in lines 223-225 in the revised manuscript.

[Figure]

Fig. 1. 2-4 month lagged correlations between the SSTA over the North Atlantic and the AMOC at 26°N during 1982-2014, with the AMOC leading (a-d, as labelled) i) in the assimilation experiment (same as Fig. 5 in the manuscript) and ii) for the 2-4 month lead time ensemble mean.

**2) The main concern I have with the paper is the interpretation of the seasonal SST skill results in terms of D16's mechanism for the strong and weak AMOC phases (section 3.3). I find the discussion insufficient to conclude whether the SST skill results are consistent D16's mechanism.**

Thanks for this remark, we agree. To make our discussion more satisfactory, we extended this discussion in the updated manuscript and highlighted to what extent D16's mechanism can explain SST predictive skill in the model. In particular, we added a new figure (Fig. 9) of area average skill following strong and weak AMOC phases across season for two regions in the North Atlantic (as in Fig. 4.a) that illustrates how significantly different skill estimates can be expected after strong and weak AMOC phases in both regions during summer, in line with D16's mechanism. We also improved this discussion to show more clearly how D16's mechanism may be hampered by atmospheric contributions to SST variability in the other seasons.

**Comments by line:**
**Line 16: 'tropical seasonal SST anomalies' ->'Seasonal SST anomalies in the tropics'?**

We agree and changed as requested. *'Seasonal SST anomalies (SSTAs) in the tropics have been linked to the intensity and genesis of tropical cyclones and heatwaves'.*

**Line 21: Could explain briefly D16's mechanism to make the paper self sustained, having only to refer to D16 for specific details.**

Agreed. To make the manuscript more self-sustained, we changed as follows: *'Here, we examine the seesaw mechanism proposed by Duchez 2016 (henceforth D16), which links the transition in strength of the Atlantic Meridional Overturning Circulation (AMOC) at 26N and North Atlantic SSTs on monthly time scales. '*

**Line 21: 'links transition' -> 'links the transition'?**

We modified this sentence to reflect this suggestion, see also previous comment.

**Lines 29-32: These sentences could be made clearer to make it easier to follow.**

We rewrote the sentences as follows, to improve clarity: *'The AMOC is estimated to transfer about 1.3 PW ($10^{15}$ W) of heat northwards at 26N (Johns et al., 2001). This heat transport, however, shows little meridional coherence at seasonal to interannual timescales (Bingham et al., 2007; Hirschi et al., 2007). Through local convergence or divergence of ocean heat transport (OHT, e.g. Cunningham et al. (2013); Borchert et al. (2018)), AMOC fluctuations could therefore influence the seasonal to interannual predictability of SST. The SST response to AMOC results in recurring large-scale patterns, generally known as AMOC fingerprints (Zhang, 2008).'*

**Lines 41-46: This paragraph could also be made clearer. The authors could explain what is meant by 'incorporating known physical mechanisms into their seasonal prediction analysis'. From the latter sentences I infer it means through the initial conditions? Perhaps the physical mechanisms could be briefly mentioned.**

We agree that this paragraph could have been clearer, changing the sentences as follows: *'Recent studies have found improved hindcast skill in the North Atlantic region after considering known physical mechanisms into their seasonal prediction analysis. Mechanisms were invoked in two possible ways: by identifying and explaining times of low and high skill, including precursors of high skill, as so-called* windows of opportunity *(Borchert et al., 2018, Mariotti et al. 2020); or by establishing physical mechanisms in the hindcast ensemble by sub-selecting ensemble members that meet certain physical criteria, thus filtering atmospheric noise in the ensemble (Dobrynin et al., 2018, Neddermann et al., 2018). The present study focuses on oceanic processes that are arguably less noisy than atmospheric dynamics (Gulev et al. 2013).*

**Line 47: The term 'potential predictability' could be briefly explained. Does it refer to the ability of the model to predict itself?**

We refer to 'SST potential predictability' as the prediction skill of SST which quantifies the fraction of long-term variability (signal) that may be distinguished from the internally generated natural variability (noise), which is unpredictable. Therefore potential predictability of SST refers to the maximum prediction skill of SST, as a function of signal-to-noise ratio, which may improve if reducing variance of systematic error and variance of noise. To address the reviewer comment, we changed the main text as follows: *'In particular, seasonal SST potential predictability, i.e. the fraction of long-term variability that may be distinguished from the internally generated natural variability, was shown to improve for better represented ocean initial states in the tropical Pacific boreal winter (Alessandri et al., 2010), and in parts of the Atlantic (Balmaseda et al., 2013).'*

**Line 49: Does 'initialised coupled simulations' refer to a set of hindcast (retrospective forecasts)? If so I would use this term to be more precise. The authors could briefly explain the evaluation technique in Borchert et al. (2018) to make the paper self sustained.**

Yes, 'initialised coupled simulations' refers to a set of hindcasts (retrospective forecasts). We changed the sentence to *'Analysing an ensemble of yearly initialised hindcasts with MPI-ESM-LR covering 1901-2010, Borchert et al., 2018, 2019 showed that the AMOC at 50°N influences the SST variability and predictability for several years, with higher skill after years of strong AMOC and vice versa. Borchert et al. (2018) perform a predictive skill analysis of SST conditioned to strong and weak OHT anomalies at 50°N separately, showing a robust influence of the ocean on windows of opportunity for decadal subpolar North Atlantic SST predictions.'*

**Line 71: This sentence should be written more precisely. 'full-field' initialisation refers to the initialisation method of the forecast system. While nudging refers to assimilation method used in the reconstruction where the initial conditions are taken from. The assimilation experiment is fully coupled right?**

We agree that this description needed rephrasing and changed as follows: *'Initial conditions of the hindcasts are taken from a fully-coupled assimilation experiment with MPI-ESM-MR. In the assimilation experiment, full temperature and salinity fields in the ocean component were nudged (Dobrynin et al., 2018) towards the ORA-S4 reanalysis (Balmaseda et al., 2013). Temperature, vorticity, divergence, and surface pressure in the atmosphere component were nudged towards ERA-Interim (Dee et al., 2011), and the sea ice component was nudged to NSIDC observations (Comiso, 1995).'*

**Line 98: Why is a linear trend removed?**

We detrend the model data to remove, as an idealised approach, the effect of anthropogenic greenhouse gas induced global warming from the analysis and focus on the internal variability, which is in line with Duchez et. al 2016 for observations. We modified the sentence as follows: *'This data set is deseasoned by removing the 12-month climatology obtained from the monthly data, and the linear trend is removed as an idealised approach to remove the externally forced signal from the time series and focus on internal variability.'*

**Line 108: 'Model verification for AMOC' -> 'Verification of the AMOC in the assimilation experiment' or 'Verification of the AMOC in the reconstruction'?**

We agree and renamed the section as 'Verification of the AMOC in the assimilation experiment'.

**Line 109: 'to' -> 'with', 'to' -> 'in'.**

Please see below in answer to 'Lines 109-110'. As we changed the section name as suggested in 'Line 108' to highlight the experiment used, we deleted lines 109-110.

**Lines 109-110: I don't think these sentences are necessary. As already mentioned I find the sentences unclear as the experiment analysed it not mentioned.**

Thanks. As we changed the section name as suggested in 'Line 108' to highlight the experiment used, we deleted lines 109-110.

**Table 1 is not discussed, so either comment on it or remove the table. What does 'seasonal range' refer to? Should be defined.**

We added a few more comments regarding the table: *'The opposite is found for the AMOC seasonal range, which is smaller for model (2.79 Sv against 3.90 Sv, Table 1)'.* Seasonal range here refers to the range of climatological values (peak-to-peak amplitude) for each time series.

**Line 112: To compute the anomalies, why is the annual mean of each year removed rather than the annual climatology?**

Thanks for the remark. This sentence was misleading and needed rephrasing, since throughout this manuscript we refer to anomalies computed by removing the annual climatology (as described in lines 118-120 in section 2.2). In this sentence we were referring to the grey lines in Fig. 1.a-f, which represent the variability within each analysed year. For Fig. 1-a,c,e, each grey line shows anomalies w.r.t. the mean transport of a given year. To make this clearer we rephrased as follows: *'To show the spread of the annual climatology,*

*grey lines in Fig.1.a,c,e represent anomalies w.r.t. the mean transport of a given year calculated for the full time series (1979-2014), and smoothed with a 3-month running average.'*

**Lines 111-122: In this paragraph figures 1b,d and f are not referred in the text or discussed, so are they necessary? Also the figures are referred in the text with 'c.f.' when it is not necessary. The clarity of the paragraph could be improved. In lines 119-120 the 'seasonal range' is commented but no reference to table 1 is provided.**

Regarding figures 1b, d and f, we added: *'There is no relevant effect of the mean state on these findings, which is why we use anomalies from now on.'* We also corrected the use of c.f. Throughout the manuscript. Lastly, we add a reference to table 1 in the sentence mentioned: *'The opposite is found for the AMOC seasonal range, which is smaller for model (2.79 Sv against 3.90 Sv, Table 1).'*

**Figure 1: The labelling of the panels in the bottom right corner instead of in the top left may be slightly confusing, although a minor detail. Is AMOC-EKM (red line) necessary in the figure 1g?**

We include the AMOC-EKM since we use this quantity in other sections of this manuscript. Panel labelling is kept as is as we believe the current version does not significantly impact readability of the figures.

**Line 127: 'model' -> 'assimilation experiment'**

We changed as suggested. *'We find that a dipole pattern represents the influence of AMOC on Atlantic SST variability in the assimilation experiment up to 7 months in the subtropical and tropical regions, similar to D16.'*

**Lines 125-142: The paragraph structure and the clarity of this section should be improved.**

We modified the structure of paragraphs and rewrote the section.

**Lines 126-130 the authors describe the lagged correlation patterns between the AMOC and the SSTA as being the same for all lags. However, to me it seems that the pattern correlation evolves from lag 0 to month 7, as later described by the authors, which seems contradictory. I believe this should be clarified.**

To make it clear that the correlation patterns change with time, we added: *' This dominant SSTA correlation pattern evolves over time.'*

**Line 145: Could reference Fig. 2 in this sentence.**

We modified as follows: *'We now analyse the impact of AMOC on SSTs over 36 years, to assess the consistency of the previous results (Fig.2) over a longer period.'*

**Line 152: 'not shown', could show in the supplementary material.**

We included one animation as supplementary material to illustrate this time dependency.

**Line 155: The first sentence seems unnecessary.**

Thanks for the suggestion. We removed the sentence and now refer to Fig.4: *'To further explore the sensitivity of AMOC fingerprints to the length of the observational window used, we show the AMOC-SST relationship averaged over two regions comprising the dipole lobes (Fig.4).'*

**Lines 159-164: This paragraph seems a repetition of the previous paragraphs.**

We rewrote the paragraph as follows, to clarify how this analysis adds to previously presented findings: *'Elaborating on findings based on spatial fields (Figs.2,3), we here show spatially aggregated SST variability (Fig.4). As before, AMOC fingerprints over the RAPID period are stronger than over the full time series. We find high anticorrelations up to 5-month lag over Box 1, ranging from -0.57 at 5-month to maximum magnitude of -0.69 at 2-month lag. In contrast, 180 when the full time series is considered, values drop to the order of -0.4. This results in a significant skill difference between the RAPID and the full period for lags 1-5 months, evaluated by non-overlapping uncertainty envelopes for the two correlation estimates (Fig.4c). Similarly, we find high correlation values above 0.6 up to 2-month lag over the RAPID period for Box 2, which drop to 0.24 at 5-month lag. The magnitude of correlations for Box 2 over the full time series reaches a maximum of 0.25. Correlation estimates for box 2 are significantly different between the two periods for lags 0-4 months (Fig.4c). Weakened AMOC fingerprints particularly during the 90s are likely responsible for the decline of the correlations computed for the full time series.'*

**Lines 155-159: Related to a previous comment, a limitation with the definition of box 2 (Fig. 4) is that the positive correlation at later lags shifts towards the Eastern North Atlantic which is not included in the box. This indicates that the lag correlation patterns are not constant.**

We agree, the lag correlation patterns do move towards east and thus out of Box 2. This is to us a nice illustration of how the impact of the fingerprint differs between the RAPID and

the full period. We therefore prefer to keep this definition. In comparing RAPID and 30-year period, we add the following remark to highlight the differences between the correlation patterns: *'This results in a significant skill difference between the RAPID and the full period for lags 1-5 months, evaluated by non-overlapping uncertainty envelopes for the two correlation estimates (Fig.4c).'*

**Line 166: delete 'here'.**

We changed the sentence as follows: *'To further investigate the variability of the SST dipole pattern, we analyse the role of SST seasonality.'*

**Lines 166-175: It is not clear what period is being used, I suppose that the 30-year Period?**

We added the period analysed as follows: *'Using the assimilation experiment for the period of 1982-2014, we perform correlations of the AMOC anomalies at a given month with the mean seasonal SSTA 2-4 months ahead (Fig.5).'*

**Line 174: Why atmospheric drivers? Is this the only option? In fact, this seems inconsistent with the results shown in the next section (Fig. 7m).**

By atmospheric drivers we mean air-sea heat fluxes, and not Ekman transport. We understand that this should be rephrased for clarity. It is true that AMOC and AMOC-EKM show similar correlation patterns in Fig. 7m, which probably means that EKM is not the cause. However, the ASF-SST correlations for the subtropical region in figure 6f are positive and therefore indicate an influence of the atmosphere in driving SST variability. Which explains why we raise atmospheric drivers as a possible reason for the different AMOC-SST pattern. We rephrased as follows: *'In contrast, we find that autumn and winter seasons lack a characteristic dipole pattern (Fig. 5.c,d), showing instead only a narrow region of negative correlations over the subtropics of the order of -0.2 (-0.1) for winter (autumn). The absence of a dipole pattern in autumn and winter may suggest the influence of atmospheric drivers that could potentially supersede the AMOC fingerprints during these seasons.'*

**Line 177: 'is response' -> 'is the response' or 'responds'**

We changed as requested: *'At the seasonal timescale, much of the SST variability in the North Atlantic is the response to atmospheric forcing (Deser et al., 2010)'.*

**Lines 176-213: The clarity of this section could be improved and perhaps made shorter. I believe this section could also be improved by commenting on the implications of the Results.**

Thanks. We took the reviewer's suggestion and improved the text for this section. In particular, we added more comments on the implications of the results: *'We compute correlations between cumulative ASF anomalies and SSTA for 2 and 4 months (where ASF leads) for each seasonal mean SSTA (Fig.6), thus highlighting regions and seasons where the atmosphere strongly contributes to SST variability. ASFs are defined as positive into the ocean, i.e. positive correlations with SST are interpreted as SST response to atmospheric heat fluxes, and vice versa. Consequently, significant positive correlations between cumulative ASF and SSTA indicate significant atmospheric contribution to SST changes that, should they overlap with AMOC fingerprints identified in Figure 5, indicate a role for the atmosphere in these regions that are potentially unpredictable. As such, this analysis forms an important step towards the assessment of seasonal SST predictions.'*

**Line 218: 'The hindcasts' -> 'The hindcasts initialised in'.**

We changed as requested: *'The hindcasts initialised in FEB, MAY, AUG and NOV yield 2-4 months lead time SST targets MAM, JJA, SON and DJF, respectively.'*

**Line 222: I assume that AUG and MAY refer to JJA and SON in fig. 8, if so this makes the text confusing, it would help to refer to the season like in fig. 8.**

We rephrased the sentence as follows: *'SON and JJA SSTs show the lowest ACCs over the subtropics, with the former showing particularly low values below 0.1 in large areas.'*

**Line 229: 'The role of AMOC' -> 'The role of the AMOC'.**

We changed the section title as requested.

**Line 231: What does 'separately' mean here? Is there any difference with the previous Section?**

Yes, there is a difference to the previous paragraphs. In lines 215-227 we describe the ACCs for each start date considering the whole time series (Fig.8a, d, g, j). In section 3.3.1 we calculate the ACCs separately only for years of strong (positive anomalies) and weak (negative anomalies) of the AMOC at 26°N one month before initialisation. We rephrased this bit to reduce confusion.

**Line 233: Does this mean that the start dates are divided into two sets, strong and weak AMOC?**

Yes. Please refer to the response above. We also tried to make this clear in the updated manuscript.

**Line 233 'Atlantic Meridional Variability (AMV)' -> 'Atlantic Multidecadal Variability (AMV)'?**

Thanks. Please note that while rewriting, we decided to remove this sentence from the manuscript to make the discussion more focused on the AMOC strength as the chosen criterion.

**Line 238: 'sub-' -> 'sub-tropical'**

We changed to 'subtropical': *'D16's physical mechanism suggests that via convergence (divergence) of OHT in the subtropics (tropics), a strong AMOC at 26N drives the subtropical and tropical SST variability at a maximum of 2-5 months lead time.'*

**Line 242: 'we find higher hindcast skill for DJF. . .' this statement is not so clear to me. The magnitude of the skill in some regions seems higher, but the area of skill seems smaller.**

We agree with the reviewer that this bit deserves a more nuanced presentation. We therefore changed the phrasing to reflect details.

**Lines 237-258: These paragraphs constitute some of the main results of the paper, but they seem insufficient to me and hard to follow. The interpretation of D16's physical mechanism in terms of skill does not seem very clear and I struggle to follow the interpretation of the results. Also, based on the AMOC seasonal fingerprints I would expect that D16's mechanism could perhaps explain the SST skill in spring and maybe in summer, but not autumn or winter.**

Thanks for alerting us to these issues; these paragraphs are indeed essential in conveying the key findings of the manuscript. We tried to be clearer by rewriting the paragraph as well as adding the new figure 9 and its explanation in the manuscript. In essence, the reviewer is right in expecting high skill during MAM and JJA from the variability analysis (Fig. 5). The findings, however, are actually even more nuanced than that: during SON and MAM, significant atmospheric contributions to SST variability may obstruct skillful predictions as well (Figs. 6 and 7). So the only season where skillful predictions can be expected to arise from the AMOC seesaw mechanism is JJA. This is shown in figure 9, where we illustrate the significance of the skill differences between strong and weak AMOC, and also reflected in the updated manuscript text.

**Line 241: The statement: 'For strong AMOC phases, we find higher hindcast skill for DJF, JJA and MAM SSTAs over the subtropics in comparison to ACCs considering the full period', I am not certain about this statement. Comparing the skill maps it seems that the**

**magnitude of the skill is greater in some regions but the areas with positive skill smaller. Perhaps computing maps of the difference of correlation (including the significance) could help the interpretation. Could the SST skill maps be affected by the smaller sample size when comparing the entire period and positive and negative AMOC composites?**

To better illustrate the difference in ACCs for strong and weak AMOC states, we add Fig. 9 in the manuscript, showing the difference of spatially aggregated skill estimates; please consider our response to your previous comment as well. The reviewer's concerns about the time period length are valid. In response to their suggestion, we rewrote this section refraining from quantitative statements about skill differences between strong/weak AMOC and the full period, and added a discussion of the skill differences between strong and weak AMOC (which are of similar time series length) based on the new figure 9.

[Figure]

*Fig. 9. SST ACCs against ERA-Interim at 2-4 months lead time averaged over the regions shown in Fig. 4a. a) Box 1 (10.5 ∘ - 22.5 ∘N, 22 ∘ - 55 ∘W), and b) Box 2 (28.5 ∘ - 40.5 ∘N, 40 ∘ - 70 ∘W. Black lines represent the ACCs considering the full time series (1982-2014), red lines for strong, and blue lines for weak AMOC phases. The shaded areas indicate the interquartile ranges.*

**Line 296: 'a more active ocean'? what does this refer to?**
We meant 'more active' to compare oceanic and atmospheric contributions in driving SST variability. Please note that when revising the manuscript we came to delete this sentence.

---

## Author Response (AR2)

**Please find below the responses to Referee#1 (pages 1-9), followed by the responses to Referee#2 (pages 10-12)**

**Referee 1, 2nd review**

We thank the anonymous referee for their valuable input and for highlighting weaknesses in the text. In response to their comments, we thoroughly revised the text to reflect more nuance on the results obtained. We also added analysis and refined some of the existing findings. We now find the manuscript to be substantially improved and thank the reviewer for helping us to get there. Major points addressed were as follows:
- We rewrote major parts of the manuscript to stress to what extent D16's mechanism influences the skill of SST and elaborated our interpretation and discussion. We particularly emphasise the seasonal and regional dependence of D16's mechanism.
- We include difference maps in the hindcast skill analysis to allow a more detailed comparison of the effect of D16's mechanism on the SST predictive skill in Fig. 8.
- Thanks to the reviewer's main concern on our surface heat fluxes analysis, we revisited our code and found a mistake, which resulted in a new Fig. 6 now in line with observations and facilitates the discussion of our results.
- We performed an additional analysis of ocean heat transport convergence and SST, which supports our discussion of the regional influence of D16's mechanism on SST predictability.

We provide below a point-by-point response to each comment. Please note that the referee's comments are highlighted in **bold** font, while our answers are in regular font.

**The authors have responded to by comments and criticism, and the manuscript has been improved. The new analysis of ocean heat transport convergence is useful, and the new figure 9 summarizing prediction skill is most helpful. However, the new figure 9 also highlights what is still my major concern. Throughout the text the authors highlight the skill differences consistent with their proposed mechanism, but these differences are sometimes marginal and (based on the new figure 9) do not seem to be significant. Considering that this is the key result of the paper, this apparent lack of significant differences should be better discussed. To conclude, although the manuscript has been improved, I still believe that it is in need of some substantial revisions before it is ready for publication. Detailed comments are found below.**

We thank the reviewer for this assessment. Upon revisiting the manuscript, we have made several changes (as noted above and below) and are now confident the reviewer will find the figures reflected in both presentation and discussion of our results in the text.

**Major comments:**

**1. Differences in prediction skill: As mentioned above, it is not clear from the text and figures to what extent there is a significant increase in prediction skill by considering the state of the AMOC. On l.310 it is stated that "significant skill difference (defined by non-overlapping confidence intervals) between strong and weak AMOC phases", but in my opinion the differences in skill shown in figure 9 do not reflect how the authors present the results and conclude. Comparing with the similar figure 11 in Borchert et al. (2018), it looks like the skill differences in the present study are much smaller (less significant). Hence, I think the presentation and discussion of skill differences could still be improved.**

We thank the reviewer for pointing this out. In response to this reviewer comment as well as the major comment from Referee 2, we computed skill difference maps based on AMOC phases (added to Figure 8), introduced persistence forecast as a benchmark in Figure 9, and re-considered our discussion and conclusion section as a result. We also added additional analysis on the OHT convergence issue (see major comment 2). All of these new analyses lead to a reframing of our results and the entire manuscript body, including the title.

In the Discussion and Conclusions sections specifically, we now make the point that the influence of AMOC and D16's mechanism on seasonal SST prediction skill is very nuanced (it is really only significant during summer in the subtropics), and so the value of this mechanism for predictions is very specific. We made an effort to reflect this nuance in every section of the manuscript and hope that the reviewer agrees that the results are now presented and discussed in a much more balanced fashion.

**2. Ocean heat transport convergence: The authors assess OHT convergence as the difference between two latitude bands (l.109-110, l.200-201). Why not calculate the actual convergence? From the location of Box 1 and 2 it is hard to believe that zonal transports are negligible. This should at least be checked and mentioned in the text.**

Unfortunately, due to data availability and issues of computational effort, calculating the actual heat budgets for the two boxes was not possible in the revision of this manuscript. However, we agree with the reviewer that more effort was needed to detail the influence of AMOC-related meridional heat transport and convergence on SST in boxes 1 and 2.

We approach this issue with a twofold analysis. First, we compare correlation between OHT convergence and SST variations in both boxes to an equivalent analysis using SST in the latitudinal bands covered by the OHT convergence analysis instead of the boxes (Fig. R1). If correlations are similar between the SST boxes and the corresponding latitudinal bands (which are independent of zonal transports), that would be an indication that zonal transports have a small impact on SST variations in those boxes. We find for both boxes a decrease of correlation when using the latitude bands instead of the boxes. This indicates a non-negligible influence of zonal transports, as suspected by the reviewer. That being said, the correlations remain significant (95% level) when using box 2 or the corresponding latitudinal band, which indicates that meridional heat transport and convergence contribute

significantly to SST variations in the subtropics, despite a relevant contribution from zonal transports of heat. Box 1 SST exhibits insignificant impact of heat convergence as a whole.

Second, an analysis of spatial SST correlation patterns to meridional OHT convergence (Fig. R2) supports the previous assessment. This analysis was done to find the spatial imprint of meridional heat convergence on SST. Across all seasons (but most prominently MAM and JJA, see manuscript), meridional convergence of heat in the box 2 latitudinal band explains SST variations in the box, which indicates a relevant impact of OHT convergence there, whereas OHT convergence in latitudes of box 1 does not explain a significant portion of SST variability there (in line with the assessment shown in Fig. R1). It is noteworthy that compared to the corresponding figure using AMOC instead of OHT convergence (paper Fig. 5), correlations are reduced for the box 2 latitudinal band. This indicates (as before) that zonal transports contribute to a non-negligible extent to SST there. Since correlations in Fig. R2 (right column) are mostly significant in box 2, however, we conclude that the impact of meridional heat convergence on SST in box 2 is significant.

We made an effort to reflect these new results in the manuscript, rephrasing many parts (maybe most prominently in lines 210-212), and to discuss our findings in a more nuanced fashion. We find this greatly improved the manuscript and therefore thank the reviewer for this suggestion.

[Figure]

*Fig R1. Time series of averaged SSTA and delta OHT. In the top row, SST is averaged over Box1 and Box2 as defined in Fig4a in the manuscript, while on the bottom row the average takes the latitudinal bands corresponding to each box, as for delta OHT.*

[Figure]

*Fig R2. Correlation at 3-month lag between seasonal SST and delta OHT (with the respective season as indicated by subtitles). Delta OHT is calculated with the latitude bands respective to Box 1 (southern box) for plots on the left, and Box 2 (northern box) on the right, as in Fig.4 in the manuscript.*

**3. Surface heat fluxes: The authors find negative correlations between SST and surface heat fluxes, i.e., the ocean forcing the atmosphere (l.243). However, as stated in the**

**manuscript (l.228), it is known from observations that seasonal SST anomalies are strongly linked to atmospheric forcing. Why would the model be different, and would this in any way influence your interpretation of your results?**

In pursuing this comment, we unearthed a coding error in our original scripts. Fixing that error made the resulting ASF-SST correlation plot look much more like what would be expected from observations. So the model is in general agreement with observations. Since the regions of low ASF influence continue to overlap with those where we find significant skill influence of AMOC, this change does not impact our central findings. We rewrote the entire ASF section to accommodate our "new" results, and discuss the influence of ASF more carefully in the discussion section now (e.g. lines 237-250).

**Minor comments:**

**l.3: It is not clear what "a seesaw-like mechanism" means.**

We made an effort to briefly explain the mechanism in the abstract.

**l.24: You should consider explaining the SST anomalies associated with the tripole.**

Done.

**l.37-41: Suggest to move these sentences up to l.34 (before "[Here] We evaluate...")**

Done.

**l.44: delete "as"**

Corrected.

**l.45-46: "by sub-selecting ensemble members that meet certain physical criteria, thus filtering atmospheric noise in the ensemble" -> I think this argument needs to better explained.**

This was identified as not pertinent to the story, so we deleted this bit.

**l.47-48: I don't see the logic between the first and second part of this sentence ("...which is why it focuses").**

See above.

**l.133-135: The use of smoothing is still confusing. In their reply, the authors state that "applying the low-pass filter only for plotting time series (e.g. Fig. 1, 4) , but not for any analysis of seasonal means". This is not easy to understand from the text. Are the grey lines in Fig.1 used/necessary? You could rather show some form of error bars.**

Here, the smoothing is used to highlight the seasonal cycle. This is now pointed out in the text: *'To show the spread of the annual climatology, grey lines in Fig. 1.a, c, e represent anomalies w.r.t. the mean transport of a given year calculated for the full time series (1979-2014), and smoothed with a 3-month running average to highlight the seasonal cycle.'*

**l.153: "up to 7 months" – ahead?**

Corrected.

**l.156-157: I don't think you need a new paragraph here.**

Agreed and fixed.

**l.160-161: I'm still not convinced that any pronounced displacement of correlation is seen along the northern boundary of the STG. Could you help the reader somehow by e.g., adding the mean barotropic streamfunction to one of the panels?**

Since this was a side note that is not particularly pertinent to the story and apparently trips readers, we rewrote: *'The magnitude of the correlation (anticorrelation) drops to maximum of 0.4 (minimum of -0.5) with increase in lag. With increasing time lag (5-7 months specifically), the subtropical lobe of positive correlation shows a displacement towards the east.'*

**l.177: "Hence,..." Not sure this sentence is consistent with the previous two, but rather agrees with what is stated on l. 173.**

Agreed, we rephrased.

**l.186: No need for a new paragraph here**

Changed accordingly.

**l.203-205: The authors state that in the tropical lobe (Box 1) the correlation is significant (negative), whereas in the subtropical lobe (box 2) the correlation is weakly positive.**

**However, in Fig.5 it is stated the mean correlation for Box 1 is -0.33 and +0.46 for Box 2. So I'm not sure I understand the authors interpretation here. Also, as the relation between AMOC and SST vis OHT convergence/divergence is central in the mechanism of D16, I think these correlations (or lack thereof) should be discussed in more detail (l.209-210 just states "other factors", could be elaborated).**

Thanks! This was an error based on mis-interpretation of earlier plots. The reviewer is completely correct. We rephrased the paragraph substantially to accommodate their concerns.

**l.216: "an assessment of an attribution" – unclear**

Rephrased to be clearer.

**l.217 (and elsewhere): the authors refer to other drivers than AMOC as "non-oceanic". However, this is not justified and there are also other sources of oceanic variability.**

Yes, agreed. We rephrased to be unambiguous now.

**l.238-239: Some repetition, consider rewriting.**

We considered the reviewer's suggestion, but decided to keep the repetition for the sake of making the point of atmospheric heat flux contribution to SST and the associated sign completely clear. That being said, rewriting in that paragraph unrelated to this particular reviewer comment might have alleviated some of the redundancy (lines 238-240).

**l.273-275: Some repetition in the description of skill. Consider rewriting.**

Rewritten to avoid redundancy.

**l.288-289: How does this statement resonate with Fig.9? Except for MAM in Box 1, it seems like the confidence intervals are overlapping for "strong AMOC" and "all years".**

Both figure 8 and figure 9 changed substantially in the process of revising the manuscript. Simultaneously, we rewrote the section on AMOC influence on SST predictions substantially. We hope that the reviewer will now find the text to be consistent with all figures.

**l.299: "skill differences we find agree to some extent with D16's physical mechanism" – the authors should better explain what skill differences they refer to and more specifically how this relates to D16's mechanism.**

We added some information on our skill difference pattern expectation in lines 285-287. The sentence that this comment references does not strictly exist anymore, but we made an effort to rephrase this section to be clearer, more intuitive, and logical.

**l.308: Again, how do relate this to what is shown in Fig.9? In Box 1 there is a skill decrease for "strong AMOC" versus "all years" (although overlapping).**

We realized in preparing this revision that we never formulated our expectation of skill increase/decrease connected to the D16 mechanism. In fact, a skill increase in box1 would be expected after weak AMOC due to the accumulation of heat. We made an effort to make this clearer in the current version of the manuscript. Still, the results did not support all claims made in the last version of the manuscript, so we now discuss this in a more nuanced fashion (lines 298-306).

**l.310: "only during summer a significant skill difference" – Confidence intervals seem to be overlapping for JJA. Ans what about MAM in Box 1?**

This was corrected in the rewrite.

**l.353: Better skill for MAM in box 1?**

Indeed, the reviewer is right. The skill for MAM SST does not improve in Box 1 after weak AMOC according to the D16 mechanism, and we now corrected the text as follows: *'After weak AMOC phases, we find high tropical SST hindcast skill during DJF and JJA (among which the JJA improvement in line with the D16 mechanism is significant, cf. Fig. 9), in particular over the central hurricane main development region.'* Since the skill improvement in spring does not comply with D16's mechanism (Box 2 has higher skill after weak AMOC and vice versa), we add the following lines for the sake of completeness: *'We also find enhanced prediction skill in that region during MAM, but after strong AMOC phases. Since this skill increase in spring does not fit the D16 mechanism, it is unlikely to originate from the examined mode of AMOC fluctuations, making room for different mechanisms to be explored and discussed in the future.'*

**l.399-400: Again, from Fig.9 skill improvement is only seen in JJA (but not significant?).**

In the process of refining the language in the manuscript to reflect the nuances of our findings, this was rewritten such that it (hopefully) reflects all caveats (lines 395-402).

**l.402-403: This I assume refers to only the small patch also referred to earlier in the text. The area-averaged skill is still only 0.4 (Fig. 9).**

See previous comment.

**l.407-408: I know I'm repeating myself, but the skill improvement the authors refer to is marginal for DJF and not significant for JJA.**
Thank you indeed for repeating these comments, as they pointed us to the urgency to discuss these results in a more nuanced fashion. That being said, we are going to return the favour and repeat ourselves when we respond to this comment: See previous comment.

**Figure 4: It's very hard to see the significance lines.**

We increased the colour intensity of significance lines.

**Figure 9: As in Borchert et al (2018), you should also add the persistence skill here as a comparison.**

We added a persistence baseline for comparison as requested.

**Referee 2, 2nd review**

We thank the anonymous referee for their valuable comments and suggestions. In response to both reviews, we have thoroughly revised the text to reflect more nuance on the results obtained. We also added analysis and refined some of the existing findings. We now find the manuscript to be substantially improved and thank the reviewer for helping us to get there. We highlight the major improvements in response to this reviewer's comments as follows:

- We compute and include correlation difference maps in Fig. 8, highlighting the nuance of what we are trying to achieve in this document.
- We rewrote major portions of the manuscript to make it clearer, convey subtleties more clearly, and improve the reading flow.

We took into consideration all suggestions made by the reviewer and we provide below a point-by-point response to each comment. Please note that the referee's comments are highlighted in **bold** font, while our answers are in regular font.

**Major comments:**
**The manuscript has greatly improved since the first version. My main concern with the paper is section 3.3.1. This section contains some of the main conclusions of this paper regarding the improvements in skill, however this mainly relies on visual comparison of the maps and it is not very robust. Perhaps computing maps of correlation differences including the statistical significance, as done in seasonal and decadal forecast studies, would help the clarity and would be a more robust way of determining skill improvements, providing stronger conclusions.**

We thank the reviewer for their assessment and the suggestion. We agree that including differences in skill does improve the interpretation of Fig. 8, and we added a new figure showing the difference maps and the regions where we expect to see an influence of the seesaw mechanism.

**Comments by line:**
**Line 144: AMC-EKM -> AMOC-EKM**

Changed.

**Line 153: Second 'subtropical' should be 'tropical'?**

Yes, changed to tropical.

**Lines 189-196: Check this paragraph as I believe that several sentences reference wrong panels of figure 4.**

Indeed there were wrong references to Fig. 4. We fixed them accordingly.

**Lines 203-205: In figure 4e the correlation is -0.33, while in figure 4f the correlation stated is 0.46 (although the latitudinal bands are inconsistent with the figure caption). Are these values correct? If so, how come the latter is referred to as 'weak' even though it's magnitude is greater than the former?**

Thanks. Indeed there was a mistake in the latitudinal bands caption, which we fixed. One correlation value was also incorrect (0.48 and not 0.46), and the reviewer is right in pointing out the inconsistency in the text. We rewrote the text as follows: '*This analysis shows significant negative correlation of AMOC with OHT convergence in the latitudes of the tropical lobe (Fig. 4e), showing that AMOC-related outflow of heat represents a relevant driver of heat convergence changes in the area. Further, AMOC is strongly positively correlated to OHT convergence in the latitudes of the subtropical lobe of the AMOC fingerprint, indicating a substantial impact of AMOC-related heat transport on oceanic heat convergence there.*'

**Figure 4 caption: The latitude bands in the caption for f) are different to what is indicated in figure.**

Thanks, we fixed the mistake.

**Line 258: 'less' is repeated.**

Fixed.

**Lines 288-289: Concluding that there is an improvement in ACC skill over the subtropics in the strong AMOC cases with respect to the entire time period by visually comparing the maps does not seem very evident to me, perhaps only in JJA. As indicated in my previous review comments, the magnitude of the skill greater but the regions of positive skill are smaller. Furthermore, in Fig. 9, you have shown that JJA has greater skill in strong AMOC event, but this is not the case for DJF nor MAM.**

We thank the reviewer for pointing this out. As explained above, the manuscript underwent a major rewrite, pointing out the nuance of D16's mechanism for seasonal SST predictions. In the process, we added skill difference plots (see response to major comment) and discussed the deficiencies of our mechanism (e.g. its inability to explain skill changes during

spring, autumn and winter) in more detail. We hope that the edits made to the manuscript now convey our results in a more balanced fashion and thus satisfy the reviewer.

**Lines 309-311: Figure 9 is a nice addition to the paper. It would help the discussion if this figure would include if the correlations and the differences in correlations are statistically significant. It would provide more robust results.**

We added this information to Figure 9 by highlighting significant skill in dots and insignificant skill using crosses. We believe this really improved the message of the figure!